# ORBIT - Open Recommendation Benchmark for Reproducible Research with Hidden Tests

Jingyuan He[1]    Jiongnan Liu[1]    Vishan Vishesh Oberoi[1*]    Bolin Wu[1*]
Mahima Jagadeesh Patel[1*]    Kangrui Mao[1*]    Chuning Shi[1*]
I-Ta Lee[2]    Arnold Overwijk[2]    Chenyan Xiong[1]

[1]Language Technologies Institute, Carnegie Mellon University    [2]Meta

{jingyuah, jiongnal, voberoi, bolinw}@andrew.cmu.edu
{mjagadee, karrym, chunings}@andrew.cmu.edu
{arnoldov, italee}@meta.com    cx@cs.cmu.edu

## Abstract

Recommender systems are among the most impactful AI applications, interacting with billions of users every day, guiding them to relevant products, services, or information tailored to their preferences. However, the research and development of recommender systems are hindered by existing datasets that fail to capture realistic user behaviors and inconsistent evaluation settings that lead to ambiguous conclusions. This paper introduces the **O**pen **R**ecommendation **B**enchmark for Reproducible Research with H**I**dden **T**ests (**ORBIT**), a unified benchmark for consistent and realistic evaluation of recommendation models. ORBIT offers a standardized evaluation framework of public datasets with reproducible splits and transparent settings for its public leaderboard. Additionally, ORBIT introduces a new webpage recommendation task, ClueWeb-Reco, featuring web browsing sequences from 87 million public, high-quality webpages. ClueWeb-Reco is a synthetic dataset derived from real, user-consented, and privacy-guaranteed browsing data. It aligns with modern recommendation scenarios and is reserved as the hidden test part of our leaderboard to challenge recommendation models' generalization ability. ORBIT measures 12 representative recommendation models on its public benchmark and introduces a prompted LLM baseline on the ClueWeb-Reco hidden test. Our benchmark results reflect general improvements of recommender systems on the public datasets, with variable individual performances. The results on the hidden test reveal the limitations of existing approaches in large-scale webpage recommendation and highlight the potential for improvements with LLM integrations. ORBIT benchmark, leaderboard, and codebase are available at `https://www.open-reco-bench.ai`.

## 1 Introduction

Recommender systems are among the most pervasive and influential AI applications today, enhancing user experience by delivering customized suggestions and reducing information overload from massive sources. Additionally, recommender systems are major revenue drivers for digital services and business platforms from e-commerce to social media applications. As such, recommender systems have become indispensable to both users and service providers and continue to attract significant attention from academia [1, 2, 3, 4].

However, the progress of recommendation models in real-world settings often deviates significantly from that measured using publicly available research datasets. Many datasets rely on crawled reviews

---

*Equal contribution.

39th Conference on Neural Information Processing Systems (NeurIPS 2025) Track on Datasets and Benchmarks.

and comments to approximate user action sequences, which differ substantially from other user interaction sequences in real world recommender systems (e.g., browsing and purchasing). Some are constructed without explicit user consent, raising ethical and legal concerns [5]. Even worse, the evaluation setups across recommendation research vary greatly in many perspectives: data splits, inference-time candidate pool, and metrics. As reported in many previous studies [6, 7, 8], these discrepancies make it difficult to reproduce and compare results fairly across studies, leading to ambiguity of recommender system findings and slowing the advancement in this field.

This paper introduces the **O**pen **R**ecommendation **B**enchmark for Reproducible Research with **HI**dden **T**ests (**ORBIT**), a unified benchmark designed for consistent and realistic evaluations for recommender systems. ORBIT consists of two core components that directly tackle the limitations of existing datasets and evaluation practices. First, ORBIT provides reproducible evaluation over 12 representative recommendation models across 5 widely-used public datasets using consistent data splits and metrics. Second, **ORBIT** introduces a novel webpage recommendation task on **ClueWeb-Reco**, a hidden test set derived from real U.S. browsing sequences with strong privacy safeguards. Though synthetic, ClueWeb-Reco closely mirrors real user interactions, enabling more realistic evaluation of modern recommenders than prior benchmarks. Leaderboards for both public benchmarks and the hidden test promote transparency, fair comparison, and future model advancement.

To construct ClueWeb-Reco, we first collect raw user histories in modern browsers through established human research platforms with explicit consent and carefully-designed quality control filters to remove noisy entries such as scams and inappropriate content. The collected histories are then aligned to publicly accessible documents in the ClueWeb22 dataset [9] through a semantic soft matching pipeline. This mapping process serves as a carefully crafted compromise, preserving the authentic behavioral patterns found in real browsing sequences while ensuring that the released data remains fully synthetic and privacy-safe. Our analysis shows that the matched pages remain highly relevant to the original browsing histories. Therefore, ClueWeb-Reco achieves a favorable trade-off between realism and privacy, offering the most realistic yet ethically sound data release possible.

With both the public datasets and hidden test set, ORBIT provides a comprehensive evaluation of existing recommendation models. On public benchmarks, content-based models consistently outperform traditional ID-based models by better capturing temporal dynamics and leveraging item features beyond identifiers. Despite this general trend, we also observe discrepancies in individual method performance across datasets, suggesting that training volume and data sparsity significantly influence model performance. To explore the role of Large Language Models (LLMs) in recommender systems, we introduce LLM-QueryGen, a novel baseline for the hidden ClueWeb-Reco test set that frames recommendation as retrieval via LLM-generated queries. While traditional models struggle on this realistic benchmark with large item candidate pool, LLM-QueryGen shows promising performance, highlighting LLMs' potential to capture user intent and generalize to unseen items.

To summarize, the contributions of this paper are as follows:

- **Consistent benchmark for reproducible research** We introduce ORBIT, a unified benchmark that ensures fair and standardized evaluation settings, and release a public leaderboard[2] over representative models. ORBIT provides insights into nowadays recommender systems.
- **Hidden test on real-life web browsing** We introduce ClueWeb-Reco[3], the first recommendation benchmark task that closely reflects user interest in a realistic recommendation setting constructed with user consent and privacy guarantees.
- **Holistic evaluation of recent recommender systems** The ClueWeb-Reco benchmark highlights the limitations of traditional models and reveals the strong generalization ability of LLM-based query generation approaches in handling large, diverse, unseen item pools.

## 2  Related Works

Research on recommender systems has traditionally relied on several widely used datasets, such as Amazon Review [10], Yelp [11], and MovieLens [12]. These datasets contain user reviews and/or

---

[2]All experiments, data processing, benchmark construction and maintenance for this work were conducted by our team at Carnegie Mellon University. Benchmark URL: `https://www.open-reco-bench.ai`

[3]The ClueWeb-Reco dataset was collected, stored, released, and is maintained by our team at Carnegie Mellon University. Dataset URL: `https://huggingface.co/datasets/cx-cmu/ClueWeb-Reco`

ratings collected on corresponding digital platforms and have supported a large body of work in this area. However, a major limitation is that they primarily capture *purchasing*, *commenting*, or *reviewing* behaviors, rather than the more general and frequent *viewing* behavior. Compared to *viewing*, actions like *reviewing* are extremely sparse — only observed in 1–2% of interactions, as shown in prior studies [13, 14, 15]. Moreover, such feedback often has popularity bias and fails to represent general user activity. Thus, the user-item interactions in these datasets may not truly reflect user behavior or preferences in browsing or purchasing. Furthermore, some recent datasets like PixelRec [16], Tenrec [17] and MicroLens [18] were collected without explicit user consent, raising privacy and ethics concerns.

Therefore, a key challenge to recommendation benchmarking is to curate realistic recommendation sequence data while protecting users' personally identifiable information (PII). The MSMARCO dataset in search ensures privacy by requiring that each query be issued by at least 50 users to meet legal thresholds [19]. However, this approach is unsuitable for sequential recommendation data since behavior sequences are rarely repeated across users. The TREC Conversational Assistance Track [20] addresses this by (1) manually crafting sessions from search logs (2) mapping queries in sessions to public MSMARCO queries, effectively transforming real behavior into privacy-preserving, yet realistic, sequences. These strategies provide valuable precedent for our soft matching approach in constructing realistic yet privacy-compliant evaluation data.

Other than recommendation datasets, efforts have been made to develop standardized and easy-to-use benchmarking tools. DeepRec offers a TensorFlow-based framework for early rating prediction and sequential models [21]. TorchRec [22] and EasyRec [23] focus on efficient and scalable large-scale recommendation [22]. Elliot [24] supports a variety of collaborative filtering and graph-based recommendation models, emphasizing both accuracy and novelty in evaluation metrics. RecBole [25] provides a unified environment for collaborative filtering models and sequential-based models that includes efficient data processing, standardized training and evaluation pipelines. Despite these efforts, inconsistencies in experimental setups across studies like data splits, inference-time ranking strategy (full-ranking or sampling candidates), and metrics still remain. As shown in many prior studies [6, 7, 8], different splitting methods alter the ranking of recommender systems, hindering fair comparison across studies and causing evaluation flaws.

Some studies focus on benchmarking and releasing results for recommendation models. RecBench [26] integrates large language models (LLMs) with conventional models for comparative evaluations in sequential and Click-Through Rate (CTR) prediction tasks. Meanwhile, BARS [27] stands out as the only comprehensive public leaderboard that evaluates diverse collaborative filtering models, providing standardized data processing and model reproducibility. DaisyRec [28] focuses on standardized benchmarking while optimizing settings—that influence reported performance and releases the DaisyRec 2.0 library. SCREEN [29] builds a conversational recommendation benchmark with 3 sub-tasks on 20 thousand dialogues. LLMRec [30] releases a benchmark in which LLMs are evaluated across multiple recommendation tasks like rating prediction, sequential recommendation, and direct recommendation. AgentRecBench [31] proposes an unified framework to study LLM Agent-based recommenders. However, many of the state-of-the-art LLM-based recommender are not present in any of these standard benchmarks, such as TASTE [3], HLLM [4], and LLM2Rec [32].

## 3 Public Benchmarks in ORBIT

This paper introduces **ORBIT**, a unified recommendation benchmark with standardized evaluation configurations on several public datasets and a newly-collected ClueWeb-Reco dataset based on real, up-to-date user browsing histories. This section introduces the public datasets and the evaluation strategies in the ORBIT benchmarks. The ClueWeb-Reco dataset is detailed in Section 4.

### 3.1 Public Datasets

ORBIT currently includes five public datasets across five distinct domains, covering two major recommendation scenarios: movie recommendation and e-commerce product recommendations. These domains are prevalent in academic research and has distinctive user-item interaction patterns, providing a balanced and representative testbed for model evaluation.

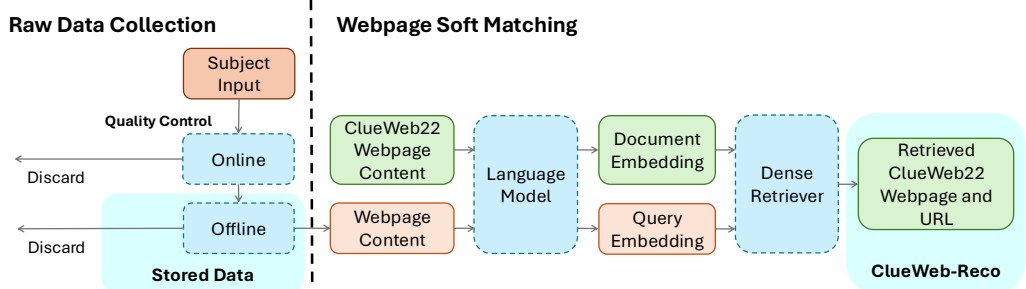

Figure 1: An illustration of the collection and processing pipeline of ClueWeb-Reco. Subject inputs that pass the two quality control checks are stored and mapped to ClueWeb22 pages through a soft-matching pipeline on the right.

**MovieLens-1M (ML-1M):** ML-1M provides 1 million user-movie interactions with explicit ratings. Its rich metadata and moderately dense user-item matrix make it an ideal dataset for evaluating long-standing recommendation performance in the movie domain.

**Amazon Reviews:** ORBIT includes four categories: *Beauty*, *Toys*, *Sports*, and *Books*, from the Amazon Reviews 2023 dataset. These datasets represent real-world e-commerce interactions and are characterized by high sparsity and long-tail item distribution, making it valuable for evaluating models under sparse data and cold-start conditions.

These five domains are deliberately selected to strike a balance between diversity and practicality. While larger datasets such as *MovieLens-20M (ML-20M)* offer interesting modeling challenges, they were excluded in this phase due to either their extensive resource demands. Besides, we will continuously include new datasets into our public benchmarks.

## 3.2 Evaluation Settings

ORBIT frames recommendation as a sequential prediction task, where the goal is to predict the next item a user will interact with, given their historical sequence. For each dataset, the model is required to select the correct next item from the entire item pool.

Specifically, we use the standard leave-one-out splitting method on all datasets to split the data into training, validation, and test sets. For each sequence grouped by user or sessions with length $n$, the first $n-2$ items are used for training models and as user history input to for validation to predict the validation target, which is the $(n-1)^{th}$ item. The $n^{th}$ item is reserved as the target for the test set while the previous $n-1$ items are given as test input.

We report two standard metrics for top-$K$ ranking: Recall@$K$ and NDCG@$K$ with $K \in \{1, 10, 50, 100\}$. Recall@$K$ measures the proportion of relevant items successfully retrieved in the top-$K$ results. Normalized Discounted Cumulative Gain (NDCG)@$K$ accounts for the rank positions of relevant items, giving higher rewards to those ranked higher. These two metrics align closely with one-relevant-item setups over top-$K$ performance. Notably, Recall@$K$ is equivalent to HitRate@$K$ in one-relevant-item setups, but is conceptually clearer as a recall-based measure.

## 4 ClueWeb-Reco: Large-scale Webpage Recommendation

This section introduces ClueWeb-Reco, a hidden test set designed to reflect recommendation models' generalization ability on the realistic webpage recommendation task. To build this dataset, we follow a two-stage pipeline as shown in Figure 1: (1) raw user browsing history collection on established human research platforms with quality control filters; (2) webpage soft matching, replacing collected webpages with relevant ones from the public ClueWeb22 corpus to fully preserve user privacy.

## 4.1 Webpage Recommendation Task

Given everyday life web usage, recommending relevant pages based on browsing behavior is both impactful and essential for diverse cases like Chrome browsing or online shopping. Therefore, we

aim to collect a hidden set over the webpage recommendation task, in which recommender systems predict the next webpage a user tends to visit based on their browsing histories. Despite its practical value in real-world applications such as browser content suggestions, building datasets for this task is challenging. Releasing real user histories risks exposing personally identifiable information (PII), while synthetic sequences generated by LLMs may not fully capture authentic user behavior [33, 34]. To overcome this, we release ClueWeb-Reco, a dataset constructed by matching consented, real browsing histories to public webpages in the ClueWeb22 corpus [9].

## 4.2 Raw User Browsing History Collection

The raw user browsing histories in ClueWeb-Reco dataset are collected via Amazon Mechanical Turk and Prolific.co under an exempt protocol approved by the Institutional Review Board of Carnegie Mellon University. Specifically, we ask users (referred to as subjects) to directly submit their personal browsing histories. All subjects are at least 18 years old and are located in the United States of America. The demographic distribution of the participants is detailed in Appendix A.1.

**Data collection** We acquire subjects' online consent to collect their data and to release a fully de-identified version of the collected data for research purposes, as illustrated in Appendix A.2. We state that by clicking the "agree" button in our study interface, subjects agree to give their consent to participate. Once consent is given, subjects are guided to the browsing history submission interface, where we provide instructions to clarify the submission process and to encourage subjects to submit data without PII. The submission instructions and interface are shown in Appendix A.4.

The raw user browsing history collection took 5 weeks, with 2 weeks of trial collection to determine strategies to facilitate high-quality submissions. 1,747 subjects give their consent to participate in the study (regardless of their submission status). Each submission that passes the online quality control discussed below is compensated $0.4 as detailed in Appendix A.3.

**Quality control** To avoid noisy and toxic information in the collected data, we apply several filters during both the online data collection stage and the offline data processing stage. As detailed in Appendix A.5, online quality control removes scams or badly formatted data, whereas offline quality control removes inappropriate and non-informative data. We collected a total of 41,760 browsing records (URL-timpstamp pairs) from 2682 raw submissions that pass the online quality control. The offline quality control stage removes 70% of the data, leaving 1024 sessions (sequences) of 12,282 browsing records (URL-timestamp pairs). As Figure 2d shows, the average sequence length (the number of browsing records) it has is 11.99, while the longest sequence has 137 browsing records.

## 4.3 Webpage Soft Matching

Even with explicit consent and careful guidelines for submissions free of personally identifiable information (PII), the risk of unintentional privacy leakage remains. For example, a subject may submit webpages related to local businesses, school application materials, or course-specific resources. When aggregated, these seemingly innocuous signals can inadvertently reveal subjects' PII [35].

To mitigate such risks and to mimic real-world webpage recommendation setup of recommending from a web corpus in terms of browsing sequence behavior, we replace each valid URL in the raw collected dataset with its most similar page in the English subset of ClueWeb22-B (ClueWeb22-B EN) [9], a large-scale information-rich web corpus. This mapping transforms private user histories into public websites to remove any PII or sensitive content, aligning with the privacy-preserving practices used in prior work such as the synthesis of MSMARCO conversational [19, 20].

To perform this transformation, we apply a semantic soft matching pipeline based on retrieval. Specifically, for each collected URL, we identify its most relevant document in ClueWeb22-B EN by computing semantic similarity using dense embeddings. As illustrated in Figure 1, the title and cleaned content of the ClueWeb corpus and the scraped content of the collected webpages are encoded using the MiniCPM-Embedding-Light model [36] into dense vectors. Then we build a dense retrieval index over the 87 million ClueWeb22-B EN pages using DiskANN [37], a state-of-the-art Approximate Nearest Neighbor Search (ANNS) method. For each collected URL, we search over this index to find the ClueWeb page with the highest semantic similarity. A higher score indicates stronger relevance, allowing us to select the closest public proxy for each private browsing URL.

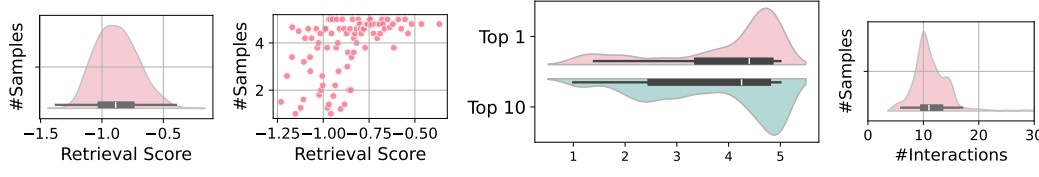

(a) Retrieval score    (b) Annotated relevance    (c) Annotated relevance vs. retrieval    (d) Session length

Figure 2: Subfigure 2a illustrates the distribution of the embedding retrieval scores between collected webpages and retrieved webpages. Subfigure 2b illustrates the average human-annotated relevance label (1-5) of each quantile of the ascending retrieval scores. Subfigure 2c illustrates the distribution of annotated relevance labels upon mapping created from different retrieval candidates. Subfigure 2d illustrates the distribution of the number of interactions in sessions of ClueWeb-Reco.

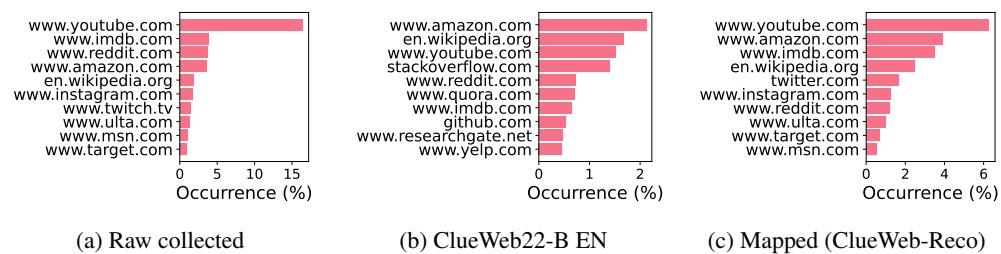

(a) Raw collected            (b) ClueWeb22-B EN          (c) Mapped (ClueWeb-Reco)

Figure 3: Top domain distribution before and after soft-matching. Subfigures 3a, 3b, and 3c illustrate the distribution of top-10 domains of the raw collected webpages, randomly sampled ClueWeb22-B EN webpages, and the mapped webpages in ClueWeb-Reco after soft-matching process, respectively.

Specifically, 11.07% of the collected URLs have an exact hit in ClueWeb22-B EN after normalization, which demonstrates ClueWeb22's good coverage of U.S. residents' browsing records. During the soft-matching process, we remove all these exact hits: if the Top-1 retrieved webpage has the same URL as the collected webpage, we use the next probably retrieval candidate (Top-2) as the mapped representation of the collected webpage. This technique ensures that the resulting ClueWeb-Reco dataset consists of fully synthetic sequences. We also ensure a one-to-one mapping during the soft-matching process: the same URL is mapped to the same page in ClueWeb corpus, whereas different URLs are mapped to different pages in ClueWeb corpus.

This final, released ClueWeb-Reco dataset consists of interaction sequences in terms of ClueWeb22-B EN document (webpage) IDs and is released under the MIT license. Note that to access the content (URLs, titles, full contents, etc) of these webpages, one must sign a license agreement to obtain the research-only ClueWeb22 dataset [9] with Carnegie Mellon University.

## 4.4 Soft Matching Quality

The soft-matching procedure balances realism and privacy by replacing collected webpages with similar public ones. We quantify the accuracy of soft-matching using retrieval scores and human annotations to assess the relevance of the mapped sequences to understand this trade-off.

**Retrieval Quality of Soft Matching.** We investigate the effectiveness of soft matching through the DiskANN-calculated similarity between embeddings, referred to as the retrieval score. A higher retrieval score indicates stronger semantic similarity between the collected webpage and its mapped counterpart in ClueWeb22-B EN. Figure 2a presents the distribution of similarity scores for all mapped pairs. It is worth mentioning that DiskANN modifies the naive inner product to calculate the relevance score, so the retrieval score can be negative. We show some matching examples in Table 1, and we can observe that the retrieval score in the range of $[-1, -0.5]$ can imply considerable semantic similarity. The distribution suggests that most collected pages have semantically close matches within ClueWeb22-B EN, supporting the viability of soft matching for representing original user real browsed webpages in a public corpus.

**Human-annotated Alignment of Soft Matching.** To further measure whether the soft-matching process preserves the user interests along the original sequences collected, 100 mappings uniformly

Table 1: The webpages from the raw collected dataset and their mapped representation in ClueWeb-Reco with embedding similarity score and averaged human-annotated relevance label.

| Webpage | | Retrieval Score | Label |
|---|---|---|---|
| **Collected** | Contemporary Music Theory - Level One: A Complete Harmony and Theory Method... *https://www.amazon.com/Contemporary-Music-Theory-Complete-Musician/dp/0793598818* | -0.5539 | 5 |
| **Retrieved** | Contemporary Music Theory - Level Three: A Complete Harmony and Theory Method... *https://www.amazon.com/Contemporary-Music-Theory-Complete-Musician/dp/0634037366* | | |
| **Collected** | Cladogram Maker \| Cladogram Generator \| Creately *https://creately.com/lp/cladogram-maker/* | -0.8271 | 4.2 |
| **Retrieved** | Organogram Template \| Online Organogram Maker \| Download and Share \| Creately *https://creately.com/lp/organogram-maker/* | | |
| **Collected** | (Sub) Culture Features - Rolling Stone *https://www.rollingstone.com/culture/culture-features/* | -0.9289 | 3.8 |
| **Retrieved** | Rolling Stone Magazine Subscription Deals *https://www.magazines.com/rolling-stone-magazine.html* | | |
| **Collected** | Bike Thief : r/UWMadison *https://www.reddit.com/r/UWMadison/comments/1k8vfcd/bike_thief/* | -1.1979 | 2.6 |
| **Retrieved** | Madison BCycle *https://madison.bcycle.com/* | | |

Table 2: Statistics of Processed Datasets Benchmarked by ORBIT

| Dataset | Dataset Information | | | | Split | | |
|---|---|---|---|---|---|---|---|
| | **#Users/Sessions** | **#Items** | **#Interactions** | **Sparsity** | **#Train** | **#Validation** | **#Test** |
| ML-1M | 6,023 | 3,044 | 995,154 | 94.57209% | 983,110 | 6,022 | 6,022 |
| Amzn-Beauty | 254 | 357 | 2,523 | 97.20439% | 2,029 | 253 | 253 |
| Amzn-Sports | 409,773 | 156,236 | 3,472,020 | 99.99458% | 2,652,476 | 409,772 | 409,772 |
| Amzn-Toys | 432,265 | 162,036 | 3,861,886 | 99.99449% | 2,997,358 | 432,264 | 432,264 |
| Amzn-Books | 776,371 | 495,064 | 9,488,297 | 99.99753% | 7,935,557 | 776,370 | 776,370 |
| ClueWeb-Reco | 1,024 | 87,208,655 | 12,282 | 99.99999% | N/A | 1024 | 1024 |

sampled from the ranked score distribution are annotated relevance label of 1 to 5, where 1 indicates completely irrelevant user interest and 5 indicates fully relevant user interest. The details of the human annotation process are discussed in Appendix A.6. The Cohen's kappa across the 5 annotators is 0.372, indicating moderate inter-annotators agreement, likely due to subjective interpretations of user intent revealed by webpage content. The average annotated relevance distribution with respect to the embedding similarity score is shown in Figure 2b. The observed trend confirms a positive correlation: higher retrieval scores generally correspond to higher annotated relevance. To validate the selection of the top-1 retrieved document, we compare its relevance scores with those of alternative mappings using the top-10 retrieval candidates. As shown in Figure 2c, top-1 matches consistently receive higher relevance scores, confirming they best reflect user intent among available candidates. Therefore, we keep the top-1 retrieved candidates to maximize the quality of soft matching to better preserve user behavioral patterns. Such design enables a small yet realistic dataset.

**Soft Matching Case study.** Some mappings between the collected webpages and the ClueWeb22-B EN webpages are shown in Table 1, with their corresponding retrieval scores and the average human-annotated relevance label. These mappings span different subjects and are all public. The top two rows show high-relevance mappings, where the topics of the collected and mapped webpages align very closely. Subsequent rows illustrate mid-relevance (label 3) and lower-relevance (label below 3) mappings, yet these still reflect key terms from the original URLs and represent thematically related interests. These examples indicate that even for the 20% of lower-relevance mappings, the soft-matching pipeline is still capable of preserving user intents.

**Soft Matching Trade-Off** As shown in the two relevance analysis and the case study above, the soft-matching process preserves the semantics of user behavior sequences. It offers a favorable trade-off, prioritizing user privacy protection while enabling the collection of a recommendation dataset that maximizes the representativeness of real-life scenarios.

## 4.5 Characteristics of ClueWeb-Reco

We evaluate the dataset's realism and diversity through domain-level distribution and sparsity analysis.

**Domain distribution analysis.** We analyze the domain distribution of the collected dataset in two steps. First, as shown in Figure 3a, the collected user browsing histories span a wide range of domains, with YouTube emerging as the most frequently visited domain. Next, Figure 3c shows the domain distribution of the mapped webpages in the final ClueWeb-Reco dataset. We observe that the top domains and their rank in ClueWeb-Reco closely mirror those in the raw collected dataset. This suggests that the soft-matching process effectively preserves domain-level characteristics, providing strong evidence of domain consistency between the original and mapped datasets.

**ClueWeb-Reco Sequence Case study.** To high-light the complexity of user behavior captured in ClueWeb-Reco, Table 3 presents an example user sequence from the validation set. The last row shows the held-out validation target for the sequence. This sequence illustrates rapid shifts in user interest across multiple topics. While some items are topically related, others diverge significantly, emphasizing the challenge of next-item prediction and the importance of modeling short-term and long-term user preferences.

Table 3: An example validation sequence in ClueWeb-Reco with its target in the last row.

| Truncated webpage title | Domain |
|---|---|
| Amazon.com: Apple MacBook Air | *www.amazon.com* |
| Apple MacBook Pro | *www.amazon.com* |
| Online Consultation \| MyHealth Clinic | *www.myhealth.ph* |
| Online Medical Consultation | *telerainmd.com* |
| Butter Pecan Cookies Recipe \| Allrecipes | *www.allrecipes.com* |
| Ne-Yo - Mad (Lyrics) - YouTube | *www.youtube.com* |
| MacBook Air 13 (2019) | *www.backmarket.com* |
| MacBook Air 13 (2017) | *www.backmarket.com* |
| How buyers can cancel an order \| eBay | *www.ebay.com.au* |

**Data sparsity.** As shown in Table 2, ClueWeb-Reco has extremely high sparsity, aligning with the real world cases when the recommendation model needs to predict the next item among massive candidates. ClueWeb-Reco also provides sessions with various numbers of interactions as in Figure 2d, covering both the cold-start and warm-start scenarios.

Built on real data submitted by users, the ClueWeb-Reco dataset can reflect real-world recommendation scenarios. Additionally, by aligning real browsing behavior with a public web corpus, it enables rigorous evaluation of model robustness with stringent privacy guarantees. The hidden nature of the test set further ensures protection against data leakage and promotes fair benchmarking.

## 5 Benchmarking Methods

This section outlines the models evaluated on the public and ClueWeb-Reco hidden test of ORBIT.

### 5.1 Public Benchmark

We consider the following representative models in the public benchmark as baselines. The detailed introduction of each specific model can be found in Appendix C.

**Sequential ID-based models** Sequential ID-based models treat user behavior as a sequence of item IDs and learnt embeddings to capture transition patterns to predict the next item, which achieves promising performance even in large-scale settings. Nevertheless, they are potentially limited in generalization to cold-start items because of their sole reliance on item ID as raw item representations. ORBIT reproduces and reports the performance of the following models: GRU4REC [38], SASRec [1], Caser [39], HGN [40], STAMP [41], FDSA [42], BERT4Rec [2], $S^3$-Rec [43], and HSTU [44].

**Sequential Content-based models:** Sequential content-based models enhance sequential recommendation by integrating item content features like product titles, tags, and descriptions into the modeling process. These models encode semantic representations of items, enabling more robust predictions in cold-start scenarios. ORBIT reproduces and reports the performance of the following models: SASRecF [45], TASTE [3], and HLLM [4]. We set the maximum history length to 50 for all models except 10 for HLLM due to our limited computational resources. The detailed implementation is included in ORBIT's codebase `https://www.open-reco-bench.ai`.

### 5.2 ClueWeb-Reco Benchmark

We consider TASTE and HLLM trained on AmazonReview-Books as content-based baselines in the ClueWeb-Reco benchmark, since content-based models exhibit generalization ability to unseen candidate representations while ID-based models do not. We additionally introduce and evaluate LLM-QueryGen baselines on ClueWeb-Reco with GPT-3.5-Turbo [46], GPT-4o, GPT-4.1 [47], Gemini-2.5-Flash [48], and Claude Sonnet 4 [49]. During LLM-QueryGen, we prompt LLMs to

Table 4: Public benchmarking results on candidate item ranking over top-10 recommended items.

| Model | ML-1M | | Amazon Beauty | | Amazon Toys | | Amazon Sports | | Amazon Books | | Average |
|---|---|---|---|---|---|---|---|---|---|---|---|
| | Recall | NDCG | Recall | NDCG | Recall | NDCG | Recall | NDCG | Recall | NDCG | NDCG |
| *ID-based* | | | | | | | | | | | |
| GRU4Rec [38] | 0.2590 | 0.1438 | 0.0157 | 0.0065 | 0.0256 | 0.0139 | 0.0228 | 0.0120 | 0.0825 | 0.0473 | 0.0447 |
| SASRec [1] | 0.2001 | 0.0967 | 0.1383 | 0.0630 | 0.0428 | 0.0209 | 0.0278 | 0.0132 | 0.0758 | 0.0384 | 0.0464 |
| Caser [39] | 0.2335 | 0.1252 | 0.0237 | 0.0082 | 0.0224 | 0.0109 | 0.0129 | 0.0061 | 0.0481 | 0.0226 | 0.0346 |
| HGN [40] | 0.0621 | 0.0300 | 0.0512 | 0.0280 | 0.0357 | 0.0182 | 0.0171 | 0.0089 | 0.0626 | 0.0289 | 0.0228 |
| STAMP [41] | 0.2217 | 0.1256 | **0.1581** | **0.0771** | 0.0323 | 0.0200 | 0.0209 | 0.0122 | 0.0600 | 0.0381 | 0.0546 |
| FDSA [42] | 0.2140 | 0.1112 | 0.0909 | 0.0504 | 0.0306 | 0.0161 | 0.0262 | 0.0138 | 0.0388 | 0.0198 | 0.0423 |
| BERT4Rec [2] | 0.3072 | 0.1820 | 0.0630 | 0.0254 | 0.0305 | 0.0165 | 0.0182 | 0.0096 | 0.0554 | 0.0325 | 0.0532 |
| $S^3$-Rec [43] | 0.3068 | **0.1899** | 0.0742 | 0.0297 | 0.0442 | 0.0214 | 0.0281 | 0.0144 | 0.0799 | 0.0398 | 0.0590 |
| HSTU [44] | **0.3236** | 0.1838 | 0.0870 | 0.0343 | 0.0401 | 0.0221 | 0.0318 | 0.0170 | 0.0672 | 0.0375 | 0.0589 |
| *Content-based* | | | | | | | | | | | |
| SASRecf [45] | 0.3045 | 0.1705 | 0.0830 | 0.0440 | 0.0345 | 0.0208 | 0.0267 | 0.0149 | 0.0709 | 0.0441 | 0.0589 |
| TASTE [3] | 0.2625 | 0.1505 | 0.0237 | 0.0122 | 0.0515 | 0.0254 | 0.0359 | 0.0183 | 0.0763 | 0.0386 | 0.0490 |
| HLLM [4] | 0.3205 | 0.1880 | 0.0079 | 0.0027 | **0.0659** | **0.0388** | **0.0443** | **0.0245** | **0.1107** | **0.0663** | **0.0641** |

Table 5: Zero-shot ClueWeb-Reco benchmarking test results on candidate item ranking.

| Model | Recall@10 | NDCG@10 | Recall@50 | NDCG@50 | Recall@100 | NDCG@100 |
|---|---|---|---|---|---|---|
| TASTE [3] | 0.0020 | 0.0015 | 0.0039 | 0.0019 | 0.0039 | 0.0019 |
| HLLM [4] | 0.0088 | 0.0041 | 0.0137 | 0.0052 | 0.0176 | 0.0059 |
| GPT-3.5-Turbo-QueryGen | 0.0088 | 0.0058 | 0.0156 | 0.0073 | 0.0254 | 0.0089 |
| GPT-4o-QueryGen | 0.0068 | 0.0027 | 0.0176 | 0.0050 | 0.0312 | 0.0072 |
| Gemini-2.5-Flash-QueryGen | 0.0068 | 0.0042 | 0.0146 | 0.0058 | 0.0264 | 0.0077 |
| GPT-4.1-QueryGen | 0.0107 | 0.0050 | 0.0195 | 0.0068 | 0.0254 | 0.0077 |
| Claude-Sonnet-4-QueryGen | 0.0068 | 0.0032 | 0.0166 | 0.0052 | 0.0215 | 0.0060 |
| DeepSeek-V3-QueryGen | 0.0127 | 0.0082 | 0.0264 | 0.0111 | 0.0371 | 0.0129 |
| Kimi-K2-QueryGen | 0.0039 | 0.0022 | 0.0156 | 0.0050 | 0.0234 | 0.0062 |
| Llama-4-Maverick-QueryGen | 0.0029 | 0.0015 | 0.0088 | 0.0028 | 0.0205 | 0.0047 |
| Qwen3-235B-QueryGen | 0.0088 | 0.0046 | 0.0234 | 0.0077 | 0.0303 | 0.0088 |

generate a query based on the interaction history verbalized by interacted webpage titles. We then retrieve the closest webpage from ClueWeb22-B EN as the prediction of LLM-QueryGen using an ANN-index. The full procedure and the prompt template we use are discussed in Appendix D.

# 6 Benchmark Results

In this section, we showcase ORBIT's public benchmark and its ClueWeb-Reco hidden test results, followed by a discussion of the performance and potential of current recommender systems.

## 6.1 Public Benchmarks

We release benchmarking results for 12 models on 5 public recommendation datasets described in Section 3.1. Table 4 presents Recall@10 and NDCG@10 results, with key observations below:

(1) We witness a consistent performance gain in sequential-based ID-based models through their evolution from RNN-based architecture to transformer architecture due to the attention-based structure is better at discovering long-term behavior patterns and user interests.

(2) Content-based models such as SASRecf and TASTE achieve better performance in general compared with ID-based models, especially in the highly sparse datasets like Amazon Books. One reason is that content-based recommendation models can utilize more meta-information of items to build more accurate user profiles and lead to better next-item predictions.

(3) The hierarchical LLM recommendation model HLLM achieves the state-of-the-art performance overall, which shows that leveraging LLMs to build item and user representations benefits the recommendation performance. However, it has poor performance in Amazon Beauty as the small training set is insufficient for tuning the billion-scale architecture.

### 6.2 ClueWeb-Reco Hidden Test

The ClueWeb-Reco hidden test result is shown in Table 5. We have the following observations:

(1) Among content-based baselines trained in the same vertical domain, HLLM generalizes much better than TASTE, likely due to its stronger LLM backbone and hierarchical architecture, which more effectively capture contextual signals.

(2) The proposed LLM-QueryGen baselines show strong predictive performance, matching or surpassing HLLM. DeekSeek achieves the highest Recall@10 and consistently strong NDCG across all cutoffs, despite the zero-shot evaluation and the challenges of ClueWeb-Reco's broad user interactions and large candidate pool.

(3) The results also reveal that different LLMs offer distinct trade-offs between top-rank precision (NDCG@10) and deep-level recall (Recall@100), which could guide practical model selection depending on application focus.

Table 6: Example queries generated by different LLMs and their corresponding NDCG@10.

| Target Webpage Title Truncated | Model | Generated Query | NDCG@10 |
|---|---|---|---|
| Amazon.com: phone ring | Gemini-2.5-Flash | best phone ring holder | **0.3333** |
| | GPT-4o | cute and durable iPhone accessories for women | 0.0000 |
| | GPT-3.5-Turbo | unique phone accessories for girls | 0.0000 |
| | GPT-4.1-QueryGen | best phone accessories for iPhone 13 girls cute design | 0.0000 |
| | Claude-Sonnet-4-QueryGen | phone accessories bundle deals | 0.0000 |
| Last of Us 2 Timeline: How Ellie & Abby | Gemini-2.5-Flash | Last of Us fanfiction | 0.0000 |
| | GPT-4o | Predator franchise timeline and connections | 0.0000 |
| | GPT-3.5-Turbo | Predator movie series chronological order | 0.0000 |
| | GPT-4.1-QueryGen | the last of us 2 ellie and dina relationship explained | **0.3562** |
| | Claude-Sonnet-4-QueryGen | post-apocalyptic survival games like The Last of Us | 0.0000 |
| Minecraft \| 10 Medieval Build Ideas and Hacks | Gemini-2.5-Flash | minecraft epic medieval builds | 0.2891 |
| | GPT-4o | medieval-themed Minecraft build ideas | **0.3562** |
| | GPT-3.5-Turbo | medieval village building tips | 0.0000 |
| | GPT-4.1-QueryGen | minecraft medieval magic base ideas | 0.0000 |
| | Claude-Sonnet-4-QueryGen | minecraft medieval building materials guide | 0.0000 |

**LLM-QueryGen Case study** Table 6 shows example predictions made by the three LLM-QueryGen baselines over the validation set of ClueWeb-Reco. The full sequence of the example in the first row is shown in Table 7. While Gemini correctly captures the user interest on "ring holder", the queries generated by other LLMs are rather broad. In the second example, GPT-4o and GPT-3.5-Turbo mistakenly focuses on the Predator movie, which appears at the early part of the session history, instead of the latest user interest over the Last of Us TV series. In the third example, GPT-3.5-Turbo misses the main concept of the game "Minecraft", resulting in degraded performance compared to other LLMs. More examples included in Appendix F show the varying performance of the LLMs as query generators in different sequences. Yet in general, state-of-the-art LLMs show higher reliability and stability over prior LLMs on generating relevant queries to the users' interest. Overall, these examples reveal the LLM's potential to capture user interests and shed light on future research that better integrate or instruct LLMs in recommendation tasks.

## 7   Limitations

ORBIT covers limited models and datasets, with plans to incorporate more public datasets and recent LLM-based recommenders like LLM2Rec [32], Molar [50], and LLMRec [51]. As an open benchmark, its impact relies on community participation, and we invite contribution and extension. ClueWeb-Reco currently focuses on U.S. user interactions; community submissions to its hidden test will help validate data authenticity and guide future expansion toward a larger supervised dataset.

## 8   Conclusions

ORBIT provides the recommender system community with a more comprehensive evaluation strategy to better reflect model performance in the large, diverse candidate pool in the real world. We would like to call for participation in submitting model predictions to our open leaderboards to expand the model coverage of ORBIT. In the future, we aim to expand our public benchmark to cover more public recommendation datasets and continue our established data collection pipeline to construct more realistic recommendation benchmarks.

## 9 Acknowledgments

We would like to thank Lee Xiong, Tianchuan Du for their valuable feedback on the paper. We also acknowledge Meta Platforms, Inc for funding for the dataset collection and the RecBole team building and maintaining RecBole [25], which serves an important role for our model evaluation experiments.

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

# A    ClueWeb-Reco Collection

## A.1    Demography

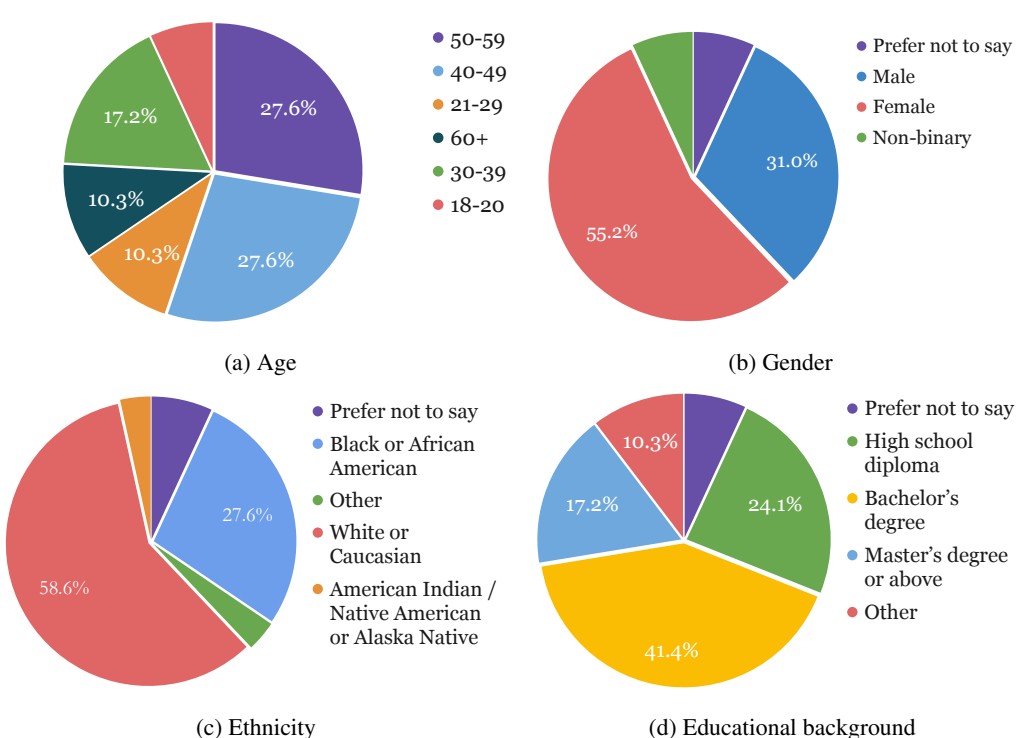

(a) Age

(b) Gender

(c) Ethnicity

(d) Educational background

Figure 4: Demography distribution of the raw collected dataset for ClueWeb-Reco.

The demographic information of the subject of ClueWeb-Reco is shown in Figure 4. The submission of demographic information is voluntary and completely anonymous. Overall, the demography information depicts an adult population who has completed high school education or more. The subject represents a population of a broad ethnic background, except for Asians. Overall, we can say that the collected data is able to represent the U.S. Internet users and their interests of browsing.

## A.2    Subject Consent and Privacy

**Subject Consent**    We provide an explicit consent form for the subject in the online entry of our study. The consent form attaches an official consent document with its main point summarized and printed in the study interface. We state that by clicking the "agree" button and continue to the study, subjects agree to give consent to participate in this study. Below we attach the Confidentiality and the Voluntary Participation sections of our consent form.

- **Risks** The risks and discomfort associated with participation in this study are no greater than those ordinarily encountered in daily life or during other online activities. The research team will not save any links non-accessible to avoid personal webpages. Yet there remains the risk of possible information identification to the research team if the submitted URLs contain any identifiable information. The collected URLs will be further processed to be fully de-identified before any possible release.

- **Confidentiality** By participating in this research, you understand and agree that Carnegie Mellon may be required to disclose your consent form, data and other personally identifiable information as required by law, regulation, subpoena or court order. Otherwise, your confidentiality will be maintained in the following manner: Your data and consent form will be kept separate. Any paper files will be stored in a secure location on Carnegie Mellon property and all digital files will be stored under Carnegie Mellon control. By participating, you understand and agree that the data and information gathered during this study may

be used by Carnegie Mellon and published and/or disclosed by Carnegie Mellon to others outside of Carnegie Mellon. However, your name, address, contact information and other direct personal identifiers will not be mentioned in any such publication or dissemination of the research data and/or results by Carnegie Mellon. Note that per regulation all research data must be kept for a minimum of 3 years. The URL data you submitted will be kept confidential. If distributed, all identifiable information in the data will be removed. The sponsor of this study (Meta Platforms, Inc.) may obtain access to the copy of de-identifiable data and research records. Third-Party Confidentiality: The study will collect your research data through your use of Prolific.co/Amazon Mechanical Turk and Vercel Neon database. These companies are not owned by CMU. The companies will have access to the research data that you produce and any identifiable information that you share with them while using their product. Please note that Carnegie Mellon does not control the Terms and Conditions of the companies or how they will use or protect any information that they collect.

- **Voluntary Participation** Your participation in this research is voluntary. You may discontinue participation at any time during the research activity. You may print a copy of this consent form for your records. By continuing to the web interface of this study, you agree that the above information has been explained to you and all your current questions have been answered. You are encouraged to ask questions about any aspect of this research study during the course of the study and in the future.

**Subject Privacy Protection** A mapping between the worker ID of Amazon Mechanical Turk or Prolific.co and this unique random identifier of a subject is stored for subject participation reward purposes. The mapping is deleted once the compensation is forwarded. Each sequence the subject submits for a unique day represents a user browsing session and is assigned a randomly generated, unique 32-character string as their identifier. All content stored and processed is under this random identifier. Any links containing personal information are inaccessible or contain error/login keywords and will not be stored in our database or will be removed from the database during post-processing.

## A.3 Subject Compensation

The equivalent $4 of cash is compensated to subjects through the human study platform the subject completed the study (Amazon Mechanical Turk and Prolific.co) for 10 valid submissions that pass the online quality control discussed in Section 4.2. A single valid submission (passes the online quality control) is prorated for $0.4. The compensation is rewarded in U.S. dollars. Depending on whether the submissions are valid and the number of submissions made, the study can take 10-30 minutes for each subject. The estimated hourly rate of the study is $8 per hour.

## A.4 ClueWeb-Reco Data Collection Procedure

To ensure the collected data reflects realistic, contemporary user behavior while protecting privacy from the start, we instruct subjects to follow these submission guidelines:

1. Submit URLs that do not contain or link to personal information;
2. Ensure the URLs link to English-language webpages;
3. Submit URLs corresponding to actual browsing activity within the past year.
4. Prefer URLs related to entertainment content where possible.

We provide subjects with two means of submissions as described below:

- **Manual Fill** Subjects are instructed to copy and paste 15 URLs and their corresponding timestamps, marked by hour and minute, one by one, into the submission box.
- **Edge Browser Export** We provide step-by-step instructions on exporting Edge browsing history. Subjects are asked to copy and paste 20 to 30 records (containing URL and visited timestamp) from the exported data into the submission box.

The two submission interfaces are shown in Figure 5. Edge Browser Export submission interface allows Edge users to submit chunks of browsing history more efficiently. We did not include a specific submission interface for Chrome because of the various formats of the exported browsing

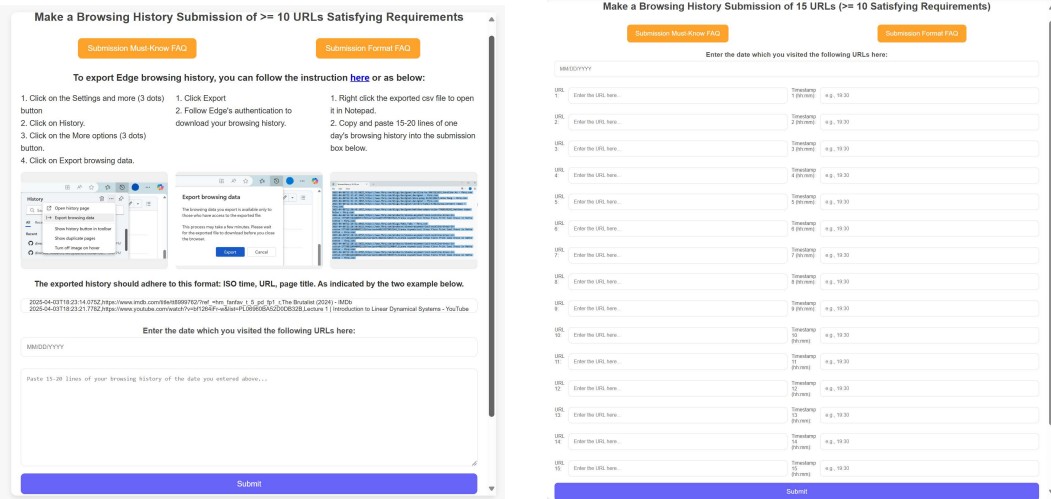

(a) Edge Browser Export Submission interface  (b) Manual Fill Submission Interface

Figure 5: The two interfaces through which subjects submit their browsing data. The Edge Browser Export contains detailed instructions on how to properly export the browsing history file from Edge.

history on different versions of Chrome. The users of Chrome and other browsers use the Manual Fill submission interface to complete the study. 45% of the subject submissions came from Edge Browser Export submission interface and the rest 55% came from the Manual Fill submission interface.

## A.5 Quality Control

Despite careful instructions, user-submitted sequences may contain noisy and toxic information. To improve the quality of user data, we apply several data filtering and selection manipulations in both the online data collection stage and the offline data processing stage.

**Online Quality Control.** When subjects submit their browsing history, we impose online checks to filter out scam or non-informative URLs and sequences. A pair of URL and its visited timestamp is considered valid to submit if it: (1) URL is a valid URL and timestamp is a valid time within 1-year's time frame in the past; (2) URL links to a publicly accessible webpage that properly handle requests; (3) URL links to neither a landing page nor a search-engine generated page upon user query; (4) is an English page; (5) passes our anti-scam checks on repeated submissions. A submission will pass and only be stored in our database if it contains at least 10 valid URL-timestamp pairs, as shown in the left half of Figure 1.

**Offline Quality Control.** After the online filtering, we manually check the browsing sequences to remove the following sequences or URL-timestamp pairs from our data storage offline to further improve the data quality:

- **Scam submission**: The most common scams are: (1) the URLs in the submission sequence has the same domains as other submission sequences in exact order; (2) the URLs in a submission sequence are from a single domain, with path names following alphabetical order; (3) the timestamps in a submission sequence are all the same or follow a fixed interval.

- **Inappropriate content**: a URL-timestamp pair is inappropriate if the URL links to a page containing violent, pornographic content, or promotes hate, harassment, or other forms of harmful behavior.

- **Non-informative**: Non-informative submissions we saw mainly fall into the following two categories: (1) the submitted URLs contain inaccessible personal content with login instructions not detected during the online-processing stage; (2) the submitted URLs link to online survey or studies; (3) the URLs in the sequences link to steps in an online game that with poor content variation.

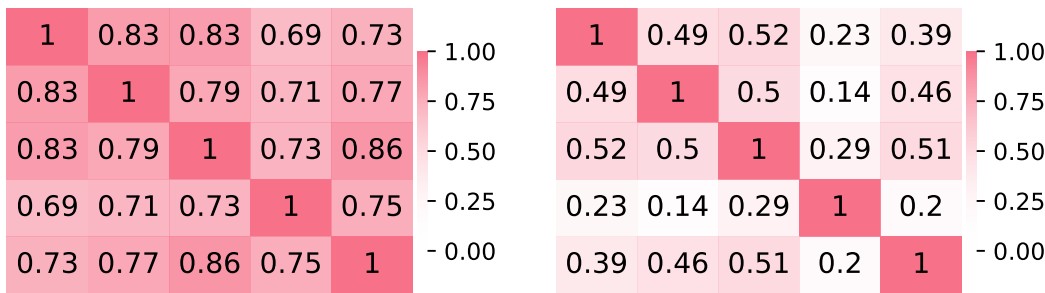



(a) Correlation of relevance labels.          (b) Cohen's kappa coefficients.



Figure 6: Correlation and Cohen's Kappa of the relevance labels given by the 5 annotators. The figures demonstrates high correlation and moderate alignment between annotators, revealing the subjective nature of human relevance annotation.

Subsequently, we represent the remaining valid URL records with the webpages they link to and employ keyword filtering on the scraped pages to further remove non-English, and non-informative webpages. Any submitted sequences with fewer than 5 URL-timestamp pairs are removed.

### A.6   ClueWeb-Reco Relevance Annotation

During the human annotation process, we sample 100 mappings uniformly from 100 buckets in the retrieval score distribution of a Top-1 retrieval scheme. We recruit 5 annotators to rate the quality of mappings on the scale of 1 to 5, where 1 indicates no relevance between the original webpage and the retrieved webpage and 5 indicates fully relevance between the original webpage and the retrieved webpage. Annotators are instructed to define webpage relevance by whether or not they depict the same user interest.

We showcase the pearson correlation of the relevance labels and the Cohen's kappa coefficients among annotations in Figure 6a. Overall, the annotators' judgments are consistent with moderate agreement upon the relevance of the raw collected webpages and the mapped webpages by the soft-matching pipeline. Consider the subjectivity over whether or not a mapping preserves user interest, this level of agreement can suggest that the relevance labels are meaningful in depicting whether or not the soft-matching pipeline is effective.

## B   ClueWeb-Reco Sequence

Table 7 shows a sequence of product browsing pages in the validation set of ClueWeb-Reco with the prediction target in the last row. Although most webpages in this session lay in Amazon, the user interest is rapidly pivoting between several loosely-connected product categories (e.g. phone case, backpack, charge adapter, phone ring holder). These categorical user interest each has varying numbers of browsing records (interactions) that are closely related, demonstrating complex, realistic user behaviors.

## C   Model Introduction

**Sequential ID-based models:**

- **GRU4REC [38]** GRU4Rec is a sequential recommendation model that uses recurrent neural networks to capture item transition patterns within a session. It models user interactions without relying on historical profiles and improves performance through data augmentation, distribution shift handling and direct item embedding prediction.
- **SASRec [1].** SASRec is a sequential recommendation model that captures the long-term semantics with a self-attention network. SASRec adaptively adjusts the weights of interacted items at each timestamp to identify the "relevant" items from a user's interaction history.

Table 7: An example validation sequence in ClueWeb-Reco with its target in the last row.

| Webpages: truncated title and URL |
| --- |
| Kawaii iPhone 13 series cases cute cheap iPhone cases various designs
*https://www.ibentoy.com/collections/iphone-case-13-series* |
| Amazon.com: vooray bag
*https://www.amazon.com/vooray-bag/s?k=vooray+bag* |
| Amazon.com: drawstring backpack with pocket
*https://www.amazon.com/drawstring-backpack-pocket/s?k=drawstring+backpack+with+pocket* |
| Amazon.com: drawstring bag pocket
*https://www.amazon.com/drawstring-bag-pocket/s?k=drawstring+bag+pocket* |
| Amazon.com: waterproof drawstring bags
*https://www.amazon.com/waterproof-drawstring-bags/s?k=waterproof+drawstring+bags* |
| Amazon.com: 20W Fast Charging Block Adapter
*https://www.amazon.com/20W-Fast-Charging-Block-Adapter/dp/B08N87FDST* |
| Amazon.com: magnetic ring holder
*https://www.amazon.com/magnetic-ring-holder/s?k=magnetic+ring+holder* |
| Amazon.com: Transparent Phone Ring Stand Holder
*https://www.amazon.com/Transparent-Phone-Ring-Stand-Holder/dp/B07ZYZQZR3* |
| Amazon.com: Jsoerpay Cell Phone Ring Holder
*https://www.amazon.com/Silitraw-Transparent-360%C2%B0Rotation-Kickstand-Compatible/dp/B0834K5QGS* |
| Spigen Style Ring Cell Phone Ring Phone Grip
*https://www.amazon.com/Spigen-Style-Holder-Phones-Tablets/dp/B0193VF09W* |
| Amazon.com: Bee Cell Phone Ring Holder with Crystal Stone
*https://www.amazon.com/Allengel-Kickstand-Compatible-Smartphone-Phone-Gold/dp/B08R3J4P5Y* |
| Amazon.com: phone ring
*https://www.amazon.com/phone-ring/s?k=phone+ring* |

- **Caser [39].** Caser is a sequential recommendation model that treats a user's interaction sequence as an "image" in time and latent spaces. It leverages horizontal and vertical convolutional filters to capture union-level and point-level sequential patterns.

- **HGN [40].** HGN captures both long-term and short-term user interests in recommendations using hierarchical gating mechanisms. It incorporates feature gating and instance gating modules to selectively pass item features at different levels, along with an item-item product module to model item relations.

- **STAMP [41].** STAMP is a sequential recommendation model that combines both long-term and short-term memory to build user profiles. It proposes a customized attention mechanism that dynamically weights these two memories to predict the next items for users.

- **FDSA [42]** FDSA is a sequential recommendation model that captures both item level and feature level transition patterns using separate self-attention blocks. It models explicit and implicit feature transitions by integrating item attributes via a vanilla attention mechanism to improve next item predictions.

- **BERT4Rec [2].** BERT4Rec uses a bidirectional transformer encoder to learn user behavior patterns for sequential recommendation tasks. Different from unidirectional models such as SASRec, BERT4Rec allows each user's historical interacted items to integrate information from both left and right sides and leads to improved recommendation accuracy across multiple benchmarks.

- **S$^3$-Rec [43].** S$^3$-Rec leverages self-supervised learning to improve sequential recommendation model performances. It introduces four self-supervised objectives, including associated attribute prediction, masked item prediction, masked attribute prediction, and segment prediction to maximize mutual information between different views of the data (items, attributes, sequences).

- **HSTU [44].** HSTU is a representative generative recommendation model that reformulates ranking and retrieval as sequential transduction tasks over unified heterogeneous feature spaces. HSTU layers employ pointwise aggregated attention to capture both long-term and recent user behaviors, allowing the model to yield strong performances while scaling linearly with sequence length and handling very large vocabularies.

**Sequential Content-based models:**

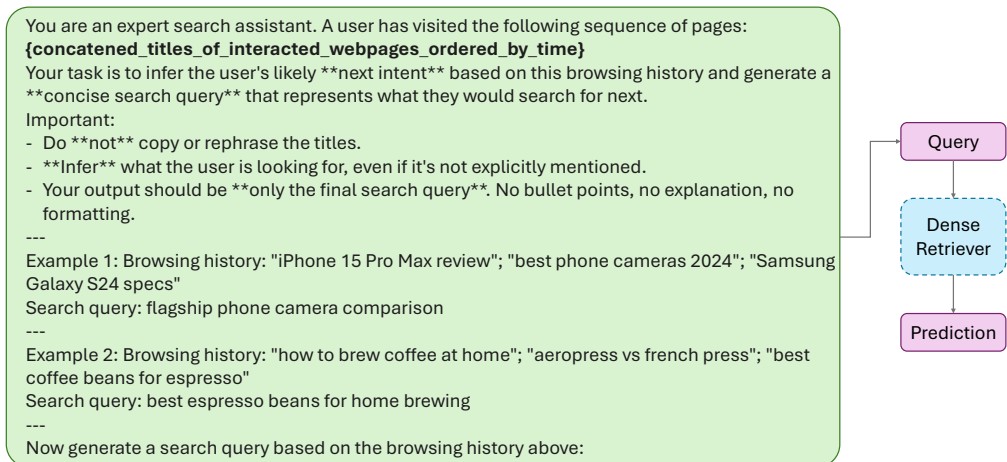

You are an expert search assistant. A user has visited the following sequence of pages:
**{concatened_titles_of_interacted_webpages_ordered_by_time}**
Your task is to infer the user's likely **next intent** based on this browsing history and generate a **concise search query** that represents what they would search for next.
Important:
- Do **not** copy or rephrase the titles.
- **Infer** what the user is looking for, even if it's not explicitly mentioned.
- Your output should be **only the final search query**. No bullet points, no explanation, no formatting.
---
Example 1: Browsing history: "iPhone 15 Pro Max review"; "best phone cameras 2024"; "Samsung Galaxy S24 specs"
Search query: flagship phone camera comparison
---
Example 2: Browsing history: "how to brew coffee at home"; "aeropress vs french press"; "best coffee beans for espresso"
Search query: best espresso beans for home brewing
---
Now generate a search query based on the browsing history above:

Query → Dense Retriever → Prediction

Figure 7: An illustration of how the LLM-QueryGen serves as a query generator that transforms the recommendation task into a retrieval pipeline and the prompt template we feed to the LLMs to obtain queries.

- **SASRecF [45].** SASRecF is an extension of the SASRec model that incorporates multimodal information such as item images, textual descriptions, and item categories into the sequential recommendation process. It extracts features using pre-trained VGG and BERT models, and combines them with item sequences through a Multimodal Attention Fusion (MAF) layer.
- **TASTE [3].** TASTE is a content-based recommendation model that verbalizes users as the concatenation of the textual representations of their historically-interacted items. TASTE leverages the embeddings of a T5 model with attention-sparsity modules for both user and item representations.
- **HLLM [4].** HLLM employs two Large Language Models (LLMs) as encoders: an item LLM to extract semantic-rich embeddings from item text, and a user LLM that models user interests based on the item LLM encoded embeddings. This design compresses detailed item content into compact vectors, improving efficiency while preserving context.

## D    LM-QueryGen Instructions

Figure 7 illustrates the LLM-QueryGen pipeline, which transforms the recommendation task as a standard dense retrieval problem. The box on the left describes the prompt we use to instruct LLMs to properly generate a query that depicts user interest, given the historically-browsed webpage sequence in a session. We represent this sequence with the concatenation of the titles of the browsed webpages. We include some examples to guide the LLM to capture user interest over shot sequence of related items as well as instructions to remove redundant, meaningless, or noisy output (e.g. a query that simply repeats the input). The right part of the same figure reveals how the LLM query generator fits in a retrieval pipeline. We encode the ClueWeb22-B EN corpus with MiniCPM-Embedding-Light [36] and build a large-scale DiskANN [37] index. We then encode the generated query with the same model and retrieve its closest document (webpage) in the ClueWeb22-B EN corpus through the index. We use the DiskANN-calculated inner product score as the retrieval score - a higher score indicates higher relevance between the query and the retrieval target. The retrieved webpage is treated as the prediction of the LM-QueryGen baseline.

## E    Experiment Evironment

We conduct our experiments using the following GPUs: NVIDIA A100 80GB PCIe, NVIDIA A100 40GB PCIe, and NVIDIA RTX A6000. The number and type of GPUs used vary depending on the size of the model.

The CPU RAM usage of different experiments varies according to the size of the dataset and the dimensions of the embeddings produced by the model.

We have include slurm job launching scripts and logs of experiments that fully describe the different resources used for each experiments in ORBIT's codebase [10].

## F   LM-QueryGen Case Studies

Table 8: Zero-shot ClueWeb-Reco benchmarking validation results on candidate item ranking.

| Model | Recall@10 | NDCG@10 | Recall@50 | NDCG@50 | Recall@100 | NDCG@100 |
|-------|-----------|---------|-----------|---------|------------|----------|
| GPT-3.5-Turbo-QueryGen | 0.0039 | 0.0016 | 0.0137 | 0.0038 | 0.0205 | 0.0049 |
| GPT-4o-QueryGen | 0.0049 | 0.0027 | 0.0098 | 0.0038 | 0.0195 | 0.0054 |
| Gemini-2.5-Flash-QueryGen | 0.0049 | 0.0017 | 0.0146 | 0.0038 | 0.0205 | 0.0047 |
| GPT-4.1-QueryGen | 0.0029 | 0.0017 | 0.0088 | 0.0031 | 0.0146 | 0.0040 |
| Claude-Sonnet-4-QueryGen | 0.0068 | 0.0034 | 0.0156 | 0.0052 | 0.0215 | 0.0062 |
| DeepSeek-V3-QueryGen | 0.0088 | 0.0043 | 0.0166 | 0.0060 | 0.0244 | 0.0073 |
| Kimi-K2-QueryGen | 0.0059 | 0.0022 | 0.0137 | 0.0039 | 0.0225 | 0.0053 |

Table 9: Example queries generated by different LLMs and their corresponding NDCG@10.

| Target Webpage Title Truncated | Model | Generated Query | NDCG@10 |
|--------------------------------|-------|-----------------|---------|
| H_ART THE BAND | Gemini-2.5-Flash | music artist discography | 0.0000 |
| | GPT-4o | music video streaming platforms | 0.0000 |
| | GPT-3.5-Turbo | H_ART THE BAND latest song | **0.6309** |
| | GPT-4.1-QueryGen | latest popular afrobeat and reggae songs 2024 | 0.0000 |
| | Claude-Sonnet-4-QueryGen | music streaming platforms comparison | 0.0000 |
| Kailah Pictures and Videos & similar | Gemini-2.5-Flash | People associated with Kailah Casillas | **0.3562** |
| | GPT-4o | Kailah Casillas personal life and relationships | 0.0000 |
| | GPT-3.5-Turbo | Kailah Casillas latest news | 0.0000 |
| | GPT-4.1-QueryGen | Kailah Casillas relationship status 2024 | 0.0000 |
| | Claude-Sonnet-4-QueryGen | Kailah Casillas dating history boyfriend | 0.0000 |
| HBO Max | Gemini-2.5-Flash | best free streaming sites | 0.0000 |
| | GPT-4o | free music streaming options 2023 | 0.0000 |
| | GPT-3.5-Turbo | queen bohemian rhapsody cover songs | 0.0000 |
| | GPT-4.1-QueryGen | how to watch music videos and live events online for free | 0.0000 |
| | Claude-Sonnet-4-QueryGen | streaming music platforms comparison | 0.0000 |
| How buyers can cancel an order \| eBay | Gemini-2.5-Flash | refurbished macbook comparison guide | 0.0000 |
| | GPT-4o | compare MacBook Air vs MacBook Pro for performance and price | 0.0000 |
| | GPT-3.5-Turbo | macbook air vs macbook pro pros and cons | 0.0000 |
| | GPT-4.1-QueryGen | best used MacBook Air or Pro for students 2024 | 0.0000 |
| | Claude-Sonnet-4-QueryGen | macbook air vs macbook pro comparison 2019 2020 | 0.0000 |

The zero-shot validation performance of the LM-QueryGen baselines is shown in Table 8. None of the LLMs demonstrates a dominant performance over the others. Rather, the ranking performance of the three LM-QueryGen baselines fluctuate across different metrics and different $K$.

Some examples of the generated queries from different model over the validation target webpage title are included in Table 9, with their corresponding NDCG@10 performance. Together with the examples in Table 6, we observe that the quality of the queries from recent GPT models (GPT-4o and GPT-4.1) is more consistent than those from GPT-3.5-Turbo, which can generate highly relevant queries as well as totally off-topic queries. The takeaway here is that state-of-the-art LLMs are more stable as a query generator for recommendation tasks, generating queries relevant to the prediction target.

## G   Asset Licenses

### G.1   Existing Assets

Amazon Review 2023 is released under MIT license. ML-1M provides a custom license[11] allowing research usage. Commercial usage and distribution is forbidden for ML-1M unless separated permission is granted. ClueWeb22 dataset is research-only and requires potential users to sign a license agreement before obtaining the dataset per their instructions[12].

---

[10]https://github.com/cxcscmu/RecSys-Benchmark

[11]https://files.grouplens.org/datasets/movielens/ml-1m-README.txt

[12]https://lemurproject.org/clueweb22/obtain.php

Our project utilizes the above datasets without data distribution or commercial interest. Each member of the team who works with ClueWeb22 dataset has signed and followed the corresponding license.

HSTU and HLLM are released under Apache-2.0 license, whereas TASTE and Recbole are released under MIT license. Both of these two licenses allow modification and distribution. Our codebase includes statements over the code source and modifications.

### G.2    New Assets

ORBIT benchmark and the newly collected ClueWeb-Reco dataset are under MIT license.

ORBIT does not use the existing assets for commercial purposes and provide data processing scripts without the raw data of the public datasets it benchmarks. For model implementation, ORBIT adapt and modify the implementation of existing open-source codebases as discussed in G.1, with the appropriate statement to credit code sources.

The information in ClueWeb-Reco dataset are released as sequences of ClueWeb IDs. Therefore, ClueWeb-Reco contains no actual ClueWeb dataset content. To access the content of these webpages, one need to sign and follow the license agreement of ClueWeb22 [9].

