# OpenReview forum: "ORBIT - Open Recommendation Benchmark for Reproducible Research with Hidden Tests"
_NeurIPS.cc/2025/Datasets_and_Benchmarks_Track — NeurIPS 2025 Datasets and Benchmarks Track poster_

### Official Review · Reviewer_N8Mx · 2025-06-11

**Rating:** 5
**Confidence:** 3

**Summary:**

This paper introduces ORBIT, a unified benchmark suite designed for reproducible, realistic evaluation of recommendation models.
ORBIT standardizes evaluation across five public datasets using consistent data splits and metrics and contributes a novel hidden test set called ClueWeb-Reco, constructed from user-consented, privacy-preserving browsing history aligned with the ClueWeb22 corpus via semantic soft-matching.
The benchmark evaluates 12 representative models and proposes LLM-QueryGen to explore generalization capabilities on ClueWeb-Reco.
Extensive experiments showcase model performance gaps across datasets and highlight the promise of LLMs for recommendation tasks.

**Dataset Code Accessibility:**

Yes

**Ethical Considerations:**

No, there are no or only very minor ethics concerns

**Final Justification:**

Thanks for the authors' detailed response, I have no further questions and believe the current score is appropriate.

However, as I am not deeply familiar with this area, I recommend that the authors and the final decision take into account the perspectives of other reviewers as well

**Limitations Weaknesses:**

1. The current version only includes datasets within movie and product domains. The initial scope could limit applicability for domains like news, music, or social media recommendations.
2. While ClueWeb-Reco is commendable in design, it contains only about 12K interactions from 1024 sessions, quite small for robust benchmarking.
3. Only TASTE and LLM-QueryGen are evaluated on ClueWeb-Reco. It would be valuable to include more content-based baselines to better characterize the task and validate the benchmark.
4. The effectiveness of LLM-QueryGen heavily depends on prompt engineering.

**Strengths Contributions:**

1. By proposing a unified benchmark with a public leaderboard and hidden test set, it fills a critical gap and can significantly improve research comparability and reproducibility: the inconsistency in evaluation problem in  RS.
2. The construction of ClueWeb-Reco is a major contribution. It creatively combines real user browsing sequences with public webpages through a privacy-preserving semantic matching process, enabling realistic yet ethical evaluation.
3. The integration of content-based and LLM-driven baselines further strengthens the relevance and modernity of the benchmark.
4. The benchmark evaluates 12 SeqRSs spanning both ID-based and content-based paradigms, giving the community a comprehensive baseline for further developments.

---

> ### Author Rebuttal · Authors · 2025-07-31
>
> Thank you for providing feedback to our work! We really appreciate them and would like to address your concerns:
>
>
> > ### **Public Dataset Coverage (Weakness 1):**
>
> We agree that more public datasets can be included to better reflect and study the behaviors of modern recommendation systems. We are currently working on the MIND-small dataset [1] and obtain the following benchmarking results as in the following table. We will continuously expand and maintain the benchmark to ensure better coverage over different recommendation behavioral patterns.
>
> | **Model**   | **Recall@10** | **NDCG@10** |
> |-------------|----------------|--------------|
> | SASRec      | 0.1866         | 0.0944       |
> | BERT4Rec    | 0.1662         | 0.0859       |
> | SASRecF     | 0.1689         | 0.0884       |
> | S3Rec       | 0.1981         | 0.1004       |
>
>
>
> > ### **ClueWeb-Reco Size and Representiveness (Weakness 2):**
>
>
> While ClueWeb-Reco includes a relatively small number of sessions, its sequence length (11.99) and sparsity (99.99%) are standard, similar to existing recommendation datasets like AmazonReview-Books, AmazonReview-Beauty (after standard 5-core filtering, as ClueWeb-Reco has sequence length longer than 5). The number of sessions of ClueWeb-Reco are large enough for conducting statistically significant experiments, and already larger than that of the 5-core filtered AmazonReview-Beauty.
>
> The highly sparse interaction patterns in ClueWeb-Reco closely reflect real-world recommendation scenarios—such as webpage suggestions in browsers like Chrome or social media feeds — where models must predict the next likely item from a vast, heterogeneous pool based on a very limited user history.
> What’s more, ClueWeb-Reco is constructed from user view/click patterns, which is dense, whereas many existing datasets are collected through user actions like reviews, which are highly sparse signals. In this sense, ClueWeb-Reco is more representative of practical recommendation challenges than many existing public datasets.
>
> ClueWeb-Reco also captures both cold-start and warm-start conditions, featuring users with as few as 5 interactions and others with over 100, as detailed in lines 269–276.
> In terms of richness, it spans a broad range of domains—including products, movies, Wikipedia, social media, and sports (Figure 3c) — and supports diverse user behavior patterns: some sequences reflect focused interests within a single domain (Table 7), while others span multiple verticals (Table 3). As shown by these examples, sequences in ClueWeb-Reco are of high quality, among the most popular pages on the Internet [2], containing features not limited to the titles we show, but also rich information like content, tags, inlinks and outlinks, etc.
>
> Thus, we believe that ClueWeb-Reco aligns with the complex heterogeneous recommendation landscape of the reality that is not covered by most of the current dataset. This sparsity and diversity mirror the complexity of real-world recommendation environments, which are often overlooked in existing datasets. While we plan to continue collecting a train set, ClueWeb-Reco already provides a strong foundation for use as a hidden test set, due to its broad coverage of realistic recommendation scenarios.
>
>
> > ### **ClueWeb-Reco Baselines (Weakness 3):**
>
> We recently add HLLM and some other LLM-QueryGen baseline evaluation over ClueWeb-Reco:
>
>
> **Table: Zero-shot ClueWeb-Reco benchmarking test results on candidate item ranking**
>
> | **Model**                  | **Recall@10** | **NDCG@10** | **Recall@50** | **NDCG@50** | **Recall@100** | **NDCG@100** |
> |---------------------------|---------------|-------------|---------------|-------------|----------------|---------------|
> | HLLM                  | 0.0088        | 0.0041      | 0.0137        | 0.0052      | 0.0176         | 0.0059        |
> | GPT-4.1-QueryGen          | 0.0107        | 0.0050      | 0.0195        | 0.0068      | 0.0254         | 0.0077        |
> | Claude-Sonnet-4-QueryGen  | 0.0068        | 0.0032      | 0.0166        | 0.0052      | 0.0215         | 0.0060        |
> | DeepSeek-V3-QueryGen      | 0.0127        | 0.0082      | 0.0264        | 0.0111      | 0.0371         | 0.0129        |
> | Kimi-K2-QueryGen          | 0.0039        | 0.0022      | 0.0156        | 0.0050      | 0.0234         | 0.0062        |
> Qwen3-235B-QueryGen | 0.0088 | 0.0046 | 0.0234 | 0.0077 | 0.0303 | 0.0088
>
>
>
> > ### **Effectiveness of LLM-QueryGen (Weakness 4):**
>
> In the ablation experiment of LLM-QueryGen over different prompts below, we does observe some variation in the performance of LLM-QueryGen baselines.
> However, despite the lower performance when tested with prompt 2, DeepSeek still out-perform Kimi and Qwen.
>
> We do want to note that it’s a common situation now that the performance of LLMs rely on prompt engineering. We encourage the academia and industry to submit custom prompts to their models to our ClueWeb-Reco to better reflect model performance.
>
>
> **Table: Ablation experiment of LLM-QueryGen with respect to differnt prompts on ClueWeb-Reco benchmarking test results**
>
>
> | **Model**                  | **Prompt 1** | | **Prompt 2** | |
> |---------------------------|---------------|-------------|---------------|-------------|
> |                           | **Recall@50** | **NDCG@50** | **Recall@50** | **NDCG@50** |
> | DeepSeek-V3-QueryGen      | 0.0264        | 0.0111      | 0.0205   | 0.0055  |
> | Kimi-K2-QueryGen          | 0.0156        | 0.0050      | 0.0146         | 0.0042        |
> | Qwen3-235B-QueryGen | 0.0234  | 0.0077 | 0.0156   | 0.0048  |
>
>
>
> --------------
> ### Reference
>
> We hope that the above discussion clarifies your concern in the comprehensiveness of ClueWeb-Reco.
> We really appreciate your suggestion over dataset coverage expansion and are working towards that direction to continuously improve ORBIT.
> Thanks again for your time and feedback!
> Please let us know if you have any further concerns.
>
> [1] Wu, F., Qiao, Y., Chen, J. H., Wu, C., Qi, T., Lian, J., ... & Zhou, M. (2020, July). Mind: A large-scale dataset for news recommendation. In Proceedings of the 58th annual meeting of the association for computational linguistics (pp. 3597-3606).
>
> [2] A. Overwijk, C. Xiong, X. Liu, C. VandenBerg, and J. Callan. ClueWeb22: 10 billion web documents with visual and semantic information. arXiv:2211.15848. 2022.

---

> > ### Comment · Reviewer_N8Mx · 2025-08-02
> > **Further Concern**
> >
> > After carefully reviewing all the reviewers’ comments and the authors’ rebuttal, I still have two concerns:
> >
> > 1. The authors describe “dense retrieval preserves semantics” as a “favorable trade-off,” but they do not quantify how much accuracy is sacrificed to achieve absolute privacy.
> >
> > 2. The dataset is drawn exclusively from U.S. users, so any conclusions based on it may not be broadly generalizable beyond that population.

---

> ### Author Response · Authors · 2025-08-02
>
> ### **Trade-off Quantification:**
>
> We have conduct several experiments in the paper to quantify relevance of mapping from user private data to public data in the paper. We use both dense retrieval scores as well as human annotations to try to show the quality of the mapping and understand the trade-off in the following manner:
>
> **Dense Retrieval Score Distribution:** Figure 2.a shows the distribution of dense retrieval scores, which are objective.
>
> **Relation between Relevance and Retrieval Scores:**
> Figure 2.b reveals the distribution of the relevance score with respect to the retrieval scores* in 100 randomly selected mappings.
> These two distributions show the relevance label distribution for all the mappings, and that the matching find relevant URLs as labeled by human.
>
> **Case Studies:**
> Table 1 shows examples of mappings and their corresponding retrieval scores and averaged human-annotated relevance score. We see that even the mapping with relatively low scores preserves signals of the original user interest (Bicycle in Madison).
>
> With these experiments in our paper, we believe the soft-matching preserves the semantics of user behavior sequences while protecting user privacy.
>
> It is quite challenging to release real user action sequences while guaranteeing privacy protection. We strongly agree with IRB and Legal reviews from our institutes, both academia and industry, that such privacy protection is necessary. We believe the soft mapping of user behavior sequences to a public dataset is one of the most effective way to curate realistic, non-simulated, user behavior data while meeting the legal and privacy requirements.
>
> In fact, this soft matching data curation technique has been employed in the construction of TREC Conversational Assistance Track  ("TREC CAsT 2019: The Conversational Assistance Track Overview"). There real user search sessions cannot be released due to privacy constraints, but the mapping to public MS MARCO queries provide a favorable trade-off of maintaining user behavior semantics and privacy protection. It formed one of the first conversational search benchmark and supported many follow up research. We hope ClueWeb-Reco and ORBIT can serve a similar role of TREC Conversational Assistance Track and provide realistic, privacy-protected, user behavior sequences for recommendation system research.
>
>
>
>
>
>
> ### **Demographic Generalization:**
>
> In the next revision of our draft, we will explicitly state that ClueWeb-Reco's generalization over other demographic regions remains unknown without further demographical investigation. We believe starting from US market is one of the common first step in constructing realistic benchmarks and how to collect more diverse dataset representing more markets is an important future research direction.
>
>
> Thank you for your time reading our work and comments. We really appreciate that and would like to further discuss if you have any further concerns!
>
>
> *: We made a typo in Figure 2.b: the y-axis should be labeled "relevance". We will fix this issue in the revision of the draft.

---

> > ### Comment · Reviewer_N8Mx · 2025-08-02
> >
> > Thank you again for your detailed response. I have no further questions and believe the current score is appropriate.
> >
> > However, as I am not deeply familiar with this area, I recommend that the authors and the final decision take into account the perspectives of other reviewers as well.

---

### Official Review · Reviewer_tBhK · 2025-07-02

**Rating:** 4
**Confidence:** 3

**Summary:**

The paper introduces ORBIT (Open Recommendation Benchmark for Reproducible Research with Hidden Tests), a unified benchmarking framework for recommendation system research that aims to provide consistent and realistic model evaluation. ORBIT consists of two main components: a standardized evaluation of 12 representative recommendation models on 5 widely-used public datasets; and a hidden test set named ClueWeb-Reco, which is a large-scale web recommendation task based on real user browsing behavior, designed to assess model generalization in open-domain settings. ClueWeb-Reco is constructed by semantically soft-matching user-submitted browsing histories with public web pages from the ClueWeb22 corpus, ensuring user privacy. Benchmark results show that content-based models generally outperform traditional ID-based approaches, and that large language models (LLMs) exhibit strong generalization when dealing with massive and diverse candidate pools.

**Dataset Code Accessibility:**

Yes

**Ethical Considerations:**

No, there are no or only very minor ethics concerns

**Final Justification:**

The authors addressed my questions, so I will keep my positive score.

**Limitations Weaknesses:**

1.	Limited dataset coverage. ORBIT currently includes datasets from only five specific domains, i.e., movies, beauty, toys, sports, and books, which may not fully represent the broader landscape of recommendation systems. Domains such as music or news may involve different user behavior patterns and data characteristics that are not captured in the benchmark.
2.	Limitations of the hidden test set. The ClueWeb-Reco dataset contains relatively sparse interaction data, which may fail to reflect the richness of real-world user–item interactions. Additionally, its single-source construction approach could limit the comprehensiveness of generalization evaluation.

**Strengths Contributions:**

1.	Standardized and consistent evaluation. ORBIT offers a unified evaluation framework covering multiple public datasets and a variety of recommendation models. This standardization ensures fair comparisons across studies and mitigates inconsistencies caused by variations in data splits, evaluation metrics, and experimental setups.
2.	Realistic user behavior data. ORBIT introduces the ClueWeb-Reco dataset, a web recommendation task based on real user browsing behavior. By semantically soft-matching user-submitted histories with public pages from ClueWeb22, it preserves user privacy while retaining realistic interaction patterns, offering a more practical basis for assessing model performance in real-world scenarios.
3.	Diverse model evaluation. ORBIT benchmarks a wide range of models, including ID-based, content-based, and LLM-based approaches. This diversity helps uncover each model type’s strengths and limitations across different datasets and tasks, providing a comprehensive view for advancing recommendation research.

---

> ### Author Rebuttal · Authors · 2025-07-31
>
> Thank you for providing feedback to our work! We really appreciate them and would like to address your concerns:
>
> > ### **Public Dataset Coverage (Weakness 1):**
>
> We agree that incorporating more public datasets would enhance the benchmark’s ability to reflect the diverse behaviors of modern recommendation systems. We are currently working on integrating the MIND-small dataset [1] and have obtained initial benchmarking results as in the following table. Moving forward, we will continue to expand and maintain ORBIT to ensure broader coverage across varying user behavior patterns and recommendation scenarios.
>
>
>
> | **Model**   | **Recall@10** | **NDCG@10** |
> |-------------|----------------|--------------|
> | SASRec      | 0.1866         | 0.0944       |
> | BERT4Rec    | 0.1662         | 0.0859       |
> | SASRecF     | 0.1689         | 0.0884       |
> | S3Rec       | 0.1981         | 0.1004       |
>
>
> > ### **ClueWeb-Reco Sparsity (Weakness 2):**
>
> The sequence length (11.99) and sparsity (99.99%) of ClueWeb-Reco are standard, similar to existing recommendation datasets like AmazonReview-Books, AmazonReview-Beauty (after standard 5-core filtering, as ClueWeb-Reco has sequence length longer than 5). The high sparsity reflects real-world recommendation settings such as webpage recommendations in Chrome or social media feeds, where models must predict the next likely item from a large and heterogeneous item pool.
> We also want to note that despite the overall high sparsity, ClueWeb-Reco includes both cold-start and warm-start conditions, with sessions ranging from as few as 5 to over 100 interactions (lines 269–276). This range allows for meaningful evaluation across interaction richness.
>
> What’s more, ClueWeb-Reco is constructed based on user view/click patterns, which is dense, whereas many existing datasets are collected through user actions like reviews, which are highly sparse signals.
> In this regard, ClueWeb-Reco is suitable as a hidden test set for diverse real-world recommendation scenarios, adding insights to current recommendation datasets.
>
>
> > ### **ClueWeb-Reco Comprehensiveness (Weakness 2):**
>
> Although ClueWeb-Reco is collected through a single interface, it is deployed on two widely used crowdworker platforms: Amazon Mechanical Turk and Prolific.co. We accept browsing history submissions from multiple browsers, including Edge and Chrome, as detailed in Appendix A.4. Voluntary demographic statistics (Appendix A.1) indicate broad coverage of U.S. internet users across various axes such as ethnicity and marital status.
>
> In terms of content richness, ClueWeb-Reco spans a wide array of domains—products, movies, Wikipedia, social media, sports, and more (Figure 3c). The sequences reflect diverse recommendation scenarios, including focused interests within a single domain (Table 7) and broad, cross-domain interests (Table 3).
> Combined with its support for both cold-start and warm-start users, ClueWeb-Reco captures the complexity and heterogeneity of real-world recommendation environments—an aspect largely missing from most existing datasets.
>
>
> We hope that the above discussion clarifies your concern in the comprehensiveness of ClueWeb-Reco.
> We really appreciate your suggestion over dataset coverage expansion and are working towards that direction to continuously improve ORBIT.
> Thanks again for your time and feedback!
> Please let us know if you have any further concerns.
>
>
> --------------
> ### Reference
>
> [1] Wu, F., Qiao, Y., Chen, J. H., Wu, C., Qi, T., Lian, J., ... & Zhou, M. (2020, July). Mind: A large-scale dataset for news recommendation. In Proceedings of the 58th annual meeting of the association for computational linguistics (pp. 3597-3606).

---

### Official Review · Reviewer_9Ade · 2025-07-02

**Rating:** 4
**Confidence:** 3

**Summary:**

This paper introduces ORBIT, a unified benchmark designed to ensure fair and standardized evaluation settings for recommender systems. One of the key contributions is the release of a public leaderboard that includes representative models. The authors also propose ClueWeb-Reco, the first recommendation benchmark task constructed from real-life web browsing data. Extensive experiments are conducted to provide a comprehensive evaluation of recent recommender systems, highlighting the benchmark’s utility and relevance.

**Dataset Code Accessibility:**

Yes

**Ethical Considerations:**

No, there are no or only very minor ethics concerns

**Final Justification:**

The authors have provided a detailed rebuttal that addresses several of my initial concerns. However, the limited number of baseline models remains an unresolved issue. Despite this, I find the core contribution meaningful and impactful, and I have raised my score to reflect this.

**Limitations Weaknesses:**

1. The dataset contains a large number of items but relatively few users, which may limit its reliability and representativeness for real-world recommendation scenarios.
2. The paper claims (line 101) that current tools suffer from inconsistencies in data splits, ranking strategies, and metrics. However, tools like Recbole already provide standardized splits, flexible metric options, and support for different ranking strategies.
3. The number of baseline models in the paper is significantly fewer than those supported by existing frameworks such as Recbole and Elliot, reducing the benchmark’s comprehensiveness.
4. Although a LLM baseline (HLLM) is included, the evaluation of LLM-based recommenders is not comprehensive, missing several recent state-of-the-art methods.

**Strengths Contributions:**

1. The authors introduce ORBIT, a unified benchmark specifically designed to support consistent and realistic evaluation of recommendation models.
2. ORBIT provides a standardized evaluation protocol, including reproducible dataset splits and transparent settings, which enhances fairness across different recommendation methods.
3. The benchmark introduces ClueWeb-Reco, a new recommendation task based on real-world web browsing behavior.

---

> ### Author Rebuttal · Authors · 2025-07-31
>
> Thank you for providing feedback to our work! We really appreciate them and would like to address your concerns:
>
>
> > ### **ClueWeb-Reco Representativeness (Weakness 1):**
>
> While ClueWeb-Reco includes a relatively small number of sessions, its sequence length (11.99) and sparsity (99.99%) are standard, similar to existing recommendation datasets like AmazonReview-Books, AmazonReview-Beauty (after standard 5-core filtering, as ClueWeb-Reco has sequence length longer than 5). The number of sessions of ClueWeb-Reco are large enough for conducting statistically significant experiments, and already larger than that of the 5-core filtered AmazonReview-Beauty.
>
> The highly sparse interaction patterns in ClueWeb-Reco closely reflect real-world recommendation scenarios—such as webpage suggestions in browsers like Chrome or social media feeds — where models must predict the next likely item from a vast, heterogeneous pool based on a very limited user history.
> What’s more, ClueWeb-Reco is constructed from user view/click patterns, which is dense, whereas many existing datasets are collected through user actions like reviews, which are highly sparse signals. In this sense, ClueWeb-Reco is more representative of practical recommendation challenges than many existing public datasets.
>
> ClueWeb-Reco captures both cold-start and warm-start conditions, featuring users with as few as 5 interactions and others with over 100, as detailed in lines 269–276. It also spans a broad range of domains—including products, movies, Wikipedia, social media, and sports (Figure 3c) — and supports diverse user behavior patterns: some sequences reflect focused interests within a single domain (Table 7), while others span multiple verticals (Table 3).
> Thus, we believe that ClueWeb-Reco aligns with the complex heterogeneous recommendation landscape of the reality that is not covered by most of the current dataset. This diversity mirrors the complexity of real-world recommendation environments, which are often overlooked in existing datasets. While we plan to continue collecting a train set, ClueWeb-Reco already provides a strong foundation for use as a hidden test set, due to its broad coverage of realistic recommendation scenarios.
>
>
> > ### **Evaluation Inconsistency Issue and Difference from RecBole (Weakness 2):**
>
> While RecBole provides flexible options for data splitting, evaluation strategies, and metrics, it does not enforce a standardized evaluation protocol. As a result, different studies using RecBole may adopt varying configurations—for example, ratio-based vs. leave-one-out splits, or full-sort vs. top-k evaluation—which undermines the comparability of reported model performance.
> ORBIT addresses this issue by establishing a fixed evaluation setting, explicitly defined and consistently applied across all benchmarked models. Built on top of RecBole’s framework, ORBIT shifts the focus from toolkit flexibility to standardized benchmarking. It offers documented hyperparameters, a consistent evaluation protocol, and public benchmarking results under these controlled settings.
>
> Moreover, ORBIT extends beyond RecBole by providing standardized splits and evaluation for the newly introduced ClueWeb-Reco dataset, including support for a hidden test set. This ensures fair comparison and reproducibility—key aspects that general-purpose toolkits like RecBole do not directly address.
>
>
> > ### **Baseline Coverage  (Weakness 3 & 4):**
>
> We acknowledge that ORBIT currently includes a smaller set of model baselines compared to frameworks like RecBole and Elliot. Expanding the set of baselines is outlined in our plan to continuously build upon and maintain the benchmark.
>
> At the same time, we would like to point out that ORBIT already covers many of the most widely used models for sequential recommendation, including SASRec and BERT4Rec. More importantly, ORBIT uniquely supports LLM-based models such as TASTE and HLLM, which are not yet integrated into RecBole or Elliot. It also introduces LLM-QueryGen baselines specifically tailored for the ClueWeb-Reco hidden test. This focus on emerging LLM-based recommenders enables ORBIT to offer insights beyond those of existing frameworks, addressing current academic and industry interest in LLM-driven personalization.
>
> We fully agree that there are additional promising LLM-baseed models — such as LLM2Rec [1] (published after Neurips submission deadline) and Molar [2] — that are not yet included in ORBIT. We are committed to expanding support for these recent advances and actively encourage the community to contribute model configurations and predictions to the benchmark. This collaborative effort will help improve model coverage and allow for more consistent and fair comparisons that better represent model performance.
>
> For ClueWeb-Reco benchmark, we have added the following baselines:
>
> **Table: Zero-shot ClueWeb-Reco benchmarking test results on candidate item ranking**
>
> | **Model**                  | **Recall@10** | **NDCG@10** | **Recall@50** | **NDCG@50** | **Recall@100** | **NDCG@100** |
> |---------------------------|---------------|-------------|---------------|-------------|----------------|---------------|
> | HLLM                  | 0.0088        | 0.0041      | 0.0137        | 0.0052      | 0.0176         | 0.0059        |
> | GPT-4.1-QueryGen          | 0.0107        | 0.0050      | 0.0195        | 0.0068      | 0.0254         | 0.0077        |
> | Claude-Sonnet-4-QueryGen  | 0.0068        | 0.0032      | 0.0166        | 0.0052      | 0.0215         | 0.0060        |
> | DeepSeek-V3-QueryGen      | 0.0127        | 0.0082      | 0.0264        | 0.0111      | 0.0371         | 0.0129        |
> | Kimi-K2-QueryGen          | 0.0039        | 0.0022      | 0.0156        | 0.0050      | 0.0234         | 0.0062        |
> Qwen3-235B-QueryGen | 0.0088 | 0.0046 | 0.0234 | 0.0077 | 0.0303 | 0.0088
>
>
> We are currently working on benchmarking the already-supported models' performance over MIND-small [3] to extend the dataset coverage of ORBIT. Here are the initial results we have so far:
>
> | **Model**   | **Recall@10** | **NDCG@10** |
> |-------------|----------------|--------------|
> | SASRec      | 0.1866         | 0.0944       |
> | BERT4Rec    | 0.1662         | 0.0859       |
> | SASRecF     | 0.1689         | 0.0884       |
> | S3Rec       | 0.1981         | 0.1004       |
>
> We hope that the above discussion can address your concern in the representativeness of ClueWeb-Reco and the contribution of ORBIT.
> We really appreciate your suggestion towards baseline coverage expansion and are working towards that direction to continuously improve ORBIT.
> Thanks again for your time and feedback! Please let us know if you have any further concerns.
>
> --------------
> ### Reference
>
> [1] He, Y., Liu, X., Zhang, A., Ma, Y., & Chua, T. S. (2025). LLM2Rec: Large Language Models Are Powerful Embedding Models for Sequential Recommendation. arXiv preprint arXiv:2506.21579.
>
> [2] Luo, Y., Qin, Q., Zhang, H., Cheng, M., Yan, R., Wang, K., & Ouyang, J. (2024). Molar: Multimodal LLMs with Collaborative Filtering Alignment for Enhanced Sequential Recommendation. arXiv preprint arXiv:2412.18176.
>
> [3]  Wu, F., Qiao, Y., Chen, J. H., Wu, C., Qi, T., Lian, J., ... & Zhou, M. (2020, July). Mind: A large-scale dataset for news recommendation. In Proceedings of the 58th annual meeting of the association for computational linguistics (pp. 3597-3606).

---

> > ### Comment · Reviewer_9Ade · 2025-08-04
> > **Comment**
> >
> > Thank you for the authors’ response, which addresses some of my concerns. I believe this work can contribute to the recommendation community, particularly in the area of model alignment evaluation. I will raise my score accordingly. However, the current set of baseline models is rather limited. I encourage the authors to incorporate more recommendation models in future work, including both traditional approaches and emerging LLM-based recommenders.

---

### Official Review · Reviewer_fV1H · 2025-07-03

**Ethics Flags:** Data privacy, copyright, and consent
**Rating:** 4
**Confidence:** 2

**Summary:**

This paper proposes an open recommendation benchmark for reproducible research. The proposed benchmark offers a standardized evaluation framework, and incorporates s new webpage recommendation task with real-world user interaction data. The experiments are extensive, covering both cold/warm-start scenarios, comparing 12 representative baselines, and including an investigation of LLMs.

**Dataset Code Accessibility:**

Yes

**Dataset Code Comments:**

Both code and dataset are publicly available. There is also a public leaderboard on the provided website.

**Ethical Comments:**

The user interaction history can release the user's sensitive information.

**Ethical Considerations:**

Yes, there are ethics concerns that require attention by the authors

**Final Justification:**

I have no more concerns. Thus, I would like to keep my positive rating.

**Limitations Weaknesses:**

1. How to deal with the data privacy issue when collecting real-world user interaction data? Why can this dataset fully preserve user privacy?
2. It is encouraged to also evaluate LLM-QuaryGen baselines on open-source LLMs, e.g., DeepSeek, LLaMA, and Qwen.

**Strengths Contributions:**

1. Reproducible research in real-world recommendation systems is indeed an interesting topic, which can cover the gap between the simulation model and real-world use.
2. The experiments are extensive, covering both cold/warm-start scenarios, comparing multiple representative baselines, and including an investigation of LLMs.
3. The authors offer a standardized evaluation framework of public datasets with reproducible splits and transparent settings for its public leaderboard, which greatly contributes to advancing research development.

---

> ### Author Rebuttal · Authors · 2025-07-31
>
> Thank you for providing feedback to our work! We really appreciate that and would like to address your concerns:
>
> > ### **Data privacy and user consent (Weakness 1):**
>
> Our newly collected ClueWeb-Reco dataset guarantees user privacy by a soft-matching pipeline similar to the construction of the TREC conversational sessions [1], which have not raised any privacy issues in the past 5 years.
> As introduced in Section 4.3 and the right portion of Figure 1, the soft-matching process maps each webpage collected to a webpage in the ClueWeb22 [2], which is a dataset protected by research-only license and comes with a privacy guarantee: all the BING-crawling webpages in ClueWeb22 are publicly reachable.
> The matched ClueWeb22 page serves as a proxy for the original collected data, making all sequences in ClueWeb-Reco fully synthetic. Since every page in the sequence comes from ClueWeb22, no private user content is included.
>
> Regarding the data privacy issue over the collection process, we follow an approved IRB protocol and explicitly obtain user consent.
> Specifically, we request subject consent over our data collection interface, and only subjects who granted consent by hitting the “Agree” button can continue to our study to submit their data (browsing history). We include the subject consent materials and discussion in Appendix A.2.
> To further safeguard privacy, we perform both online and offline quality checks to filter out inaccessible or inappropriate pages. As a result, all stored data consists of publicly accessible webpages that contain no Personally Identifiable Information (PII), even being mapping them to ClueWeb22 for inclusion in ClueWeb-Reco.
>
>
> *We want to note the quality guarantee of the soft-matching process here as well:*
>
> Since we use dense retrieval to perform this soft-matching process, the mapped sequences are able to preserve the user behaviors of the collected sequences. This is because the LLM-produced embeddings in dense retrieval maintain the semantic meaningfulness of webpages through the mapping process, as detailed in the discussion of Section 4.4.
> Table 2 shows example mappings from collected webpages to their mapped representation in the released ClueWeb-Reco. (Note that the shown collected webpages are from different sequences and consist solely of public webpages, ensuring no user browsing data is leaked.) Even though the mapped webpages are not exactly the same as the collected webpages, they exhibit high relevance, which means the mapped sequences can preserve the user behaviors in the original collected sequence. We believe this approach offers a favorable trade-off between preserving user privacy and enabling the collection of recommendation data for academic research.
>
>
>
> > ### **LLM-QueryGen Baselines (Weakness 2):**
>
> We will continuously add new LLM baselines to the ClueWeb-Reco leaderboard.
> We have added HLLM and the following LLM-QueryGen baselines so far:
>
> **Table: Zero-shot ClueWeb-Reco benchmarking test results on candidate item ranking**
>
> | **Model**                  | **Recall@10** | **NDCG@10** | **Recall@50** | **NDCG@50** | **Recall@100** | **NDCG@100** |
> |---------------------------|---------------|-------------|---------------|-------------|----------------|---------------|
> | HLLM                  | 0.0088        | 0.0041      | 0.0137        | 0.0052      | 0.0176         | 0.0059        |
> | GPT-4.1-QueryGen          | 0.0107        | 0.0050      | 0.0195        | 0.0068      | 0.0254         | 0.0077        |
> | Claude-Sonnet-4-QueryGen  | 0.0068        | 0.0032      | 0.0166        | 0.0052      | 0.0215         | 0.0060        |
> | DeepSeek-V3-QueryGen      | 0.0127        | 0.0082      | 0.0264        | 0.0111      | 0.0371         | 0.0129        |
> | Kimi-K2-QueryGen          | 0.0039        | 0.0022      | 0.0156        | 0.0050      | 0.0234         | 0.0062        |
> Qwen3-235B-QueryGen | 0.0088 | 0.0046 | 0.0234 | 0.0077 | 0.0303 | 0.0088
>
>
>
>
>
> We hope that the above discussion can clarify your concerns about data privacy. It's indeed a very important issue in publishing dataset and we will adjust the draft to make it more clear over this concern upon acceptance/released.
>
> Again, thank you for your time and feedback, and please let us know if you have any further concerns.
>
> --------------
> ### Reference
>
> [1] J. Dalton, C. Xiong, X, and J. Callan. Trec cast 2019: The conversational assistance track overview. arXiv preprint arXiv:2003.13624, 2020.407
>
> [2] A. Overwijk, C. Xiong, X. Liu, C. VandenBerg, and J. Callan. ClueWeb22: 10 billion web documents with visual and semantic information. arXiv:2211.15848. 2022.

---

> > ### Comment · Reviewer_fV1H · 2025-08-05
> >
> > Thank you for your detailed response. I have no more concerns. Thus, I would like to keep my positive rating.

---

### Note · Authors · 2025-08-13

Thank you for the feedback and we’d like to make a final remark of the author-reviewer discussion:

>**Strength of ORBIT:**
- (fV1H, 9Ade, tBhk,N8Mx) ORBIT enhances reproducibility and comparability in RS by hosting a public leaderboard supported by standard, consistent evaluation protocol and transparent settings.
- (fV1H, tBhk, N8Mx) ORBIT conducts experiments across representative ID-based, content-based, and LLM-based models.
(fV1H) ORBIT conducts extensive experiments on multiple datasets, “covering both cold/warm-start scenarios".
- (9Ade, tBhk, N8Mx) The newly collected ClueWeb-Reco as a webpage recommendation evaluation task based on real-world browsing data.
- (tBhk, N8Mx) ClueWeb-Reco “preserves user privacy while retaining realistic interaction patterns, offering a more practical basis for assessing model performance in real-world scenarios”.

>**Reviewer’s concerns and how we address them:**

ClueWeb-Reco baseline coverage:
Reviewer N8Mx and fV1H suggest we add more content-based and open-source LLM-QueryGen baselines. We provide HLLM result on ClueWeb-Reco and include 3 more popular LLM-QueryGen baselines. Both reviewers agree and have their concerns addressed.

ClueWeb-Reco privacy and comprehensiveness:
We appreciate reviewer fV1H's feedback on data privacy. We clarify ClueWeb-Reco’s synthetic nature in the abstract and improve the draft and benchmark website to highlight our adherence to the IRB protocol and user consent. The reviewer’s comment shows "no more concerns."
Regarding concerns about ClueWeb-Reco's representativeness (reviewers 9Ade, N8Mx, and tBhK), we clarified that its sequence length and sparsity align with real-world scenarios. We also noted the dataset provides enough sessions for statistically significant experiments and is derived from authentic user view/click patterns, possessing a rich structure. Reviewer 9Ade and N8Mx agree and have their concerns addressed.

Novelty of ORBIT:
We clarify ORBIT’s novelty as a benchmark, providing fixed, transparent settings and that ORBIT includes LLM-based models not already covered. After addressing this, reviewer 9Ade promises to improve rating.

>**Things we will do in our next revision:**
- Clarify the privacy guarantees of ClueWeb-Reco
- Acknowledge ClueWeb-Reco’s soft-matching process has tradeoff between real sequences and user privacy
- Acknowledge the scalability and demographic scope of ClueWeb-Reco
- Expand the public dataset and baselines coverage of ORBIT

---

### Decision · Program_Chairs · 2025-09-18

**Decision:**

Accept (poster)

**Comment:**

This paper introduces ORBIT, a unified benchmark of recommendation models. Notably, the authors introduce a new webpage recommendation task, ClueWeb-Reco.

Strengths:
- Recommendation model work is fairly under-resourced - a meaningful, realistic evaluation suite is a helpful and necessary intervention for the community to ensure reproducibility.
- As many reviewers note, the ClueWeb-Reco development process provides a useful framework for thinking through the practical challenge of preserving privacy while also generating realistic use interaction data for the purpose of recsys evaluation.

Weaknesses:
- As raised by various reviews, the trade-off between privacy preserving measures and accuracy is very unclear and unquantified
- Notably, I have concerns about the accuracy of the LLM-based embedding process - how exactly did the authors confirm that the query/document embeddings are correct? The details provided in Appendix A.6 are very unclear (the included figure does not include any axes labels or adequately descriptive captions, and is thus not intuitive to understand). From my understanding, it seems that a small number of 5 annotators manually verified a very small sample of examples and it is not clear from the included figure alone how strongly the correlation is between these manual annotations and the LLM-derived embeddings.
- Novelty: The authors need to engage in much more extensive background review, and reference the many prior relevant attempts to modernize and standardize recommendation system evaluation (ie. 9Ade mentions RecBole, but there is also BARS [1], and several others), as well as more recent attempts to extend recommendation system platforms to include LLM models [2-4]. Without a direct comparison to these prior efforts, it unclear to identify what specific contribution specifically ORBIT brings into this ecosystem.
- As tBhK raises, there is an inherent limitation to the coverage of ORBIT - comparing the scale and coverage of this benchmark, to other recommendation benchmarks would help contextualize the contributions of the paper.

Additional questions:
- Clueweb22 [5] is a large dataset; why only retrieve "similar" sessions 1:1? Why not retrieve multiple similar queries from the same doc/query embedding and expand the size of the final dataset? As  N8Mx mentions, the current size of ClueWeb-Reco is quite limited, including only 12K interactions with a small number of users. One unstated possible advantage of this approach could be to use the  ClueWeb-Reco  to  generate this kind of realistic synthetic benchmark at scale.

Overall, the value of the methodological contribution of the ClueWeb-Reco development process, and the comprehensive expansion of ORBIT to include interaction-based and conversation recommendation systems means that this paper just passes the acceptance threshold. However, I highly recommend that author revisit reviewer comments and update their paper in according to the provided feedback.


[1] https://openbenchmark.github.io/BARS/
[2] Liu, Qijiong, et al. "Benchmarking LLMs in Recommendation Tasks: A Comparative Evaluation with Conventional Recommenders." arXiv preprint arXiv:2503.05493 (2025).
[3] Liu, Junling, et al. "Llmrec: Benchmarking large language models on recommendation task." arXiv preprint arXiv:2308.12241 (2023).
[4] Lin, Dongding, et al. "SCREEN: A Benchmark for Situated Conversational Recommendation." Proceedings of the 32nd ACM International Conference on Multimedia. 2024.
[5] Overwijk, Arnold, et al. "Clueweb22: 10 billion web documents with visual and semantic information." arXiv preprint arXiv:2211.15848 (2022).

===== FINAL UPDATE FROM DB Track PCs ====

The final decision for this paper has been taken by the program chairs after consultation with the SACs. All Senior Area Chairs have ranked papers according to the feedback from the AC during the review process. We decided to leave the original meta-review to reflect the opinion of the AC in light of the initial discussions with reviewers and SAC.